# Transcriptomic profiling of skeletal muscle adaptations to exercise and inactivity

Nicolas J. Pillon [1], Brendan M. Gabriel [1], Lucile Dollet[1], Jonathon A.B. Smith [1], Laura Sardón Puig [2], Javier Botella [3], David J. Bishop [3], Anna Krook [1] & Juleen R. Zierath [1,2,4]*

The molecular mechanisms underlying the response to exercise and inactivity are not fully understood. We propose an innovative approach to profile the skeletal muscle transcriptome to exercise and inactivity using 66 published datasets. Data collected from human studies of aerobic and resistance exercise, including acute and chronic exercise training, were integrated using meta-analysis methods (www.metamex.eu). Here we use gene ontology and pathway analyses to reveal selective pathways activated by inactivity, aerobic versus resistance and acute versus chronic exercise training. We identify *NR4A3* as one of the most exercise- and inactivity-responsive genes, and establish a role for this nuclear receptor in mediating the metabolic responses to exercise-like stimuli in vitro. The meta-analysis (MetaMEx) also highlights the differential response to exercise in individuals with metabolic impairments. MetaMEx provides the most extensive dataset of skeletal muscle transcriptional responses to different modes of exercise and an online interface to readily interrogate the database.

[1] Department of Physiology and Pharmacology, Karolinska Institutet, Stockholm, Sweden. [2] Department of Molecular Medicine and Surgery, Karolinska Institutet, Stockholm, Sweden. [3] Institute for Health and Sport, Victoria University, Melbourne, Australia. [4] Novo Nordisk Foundation Center for Basic Metabolic Research, University of Copenhagen, Copenhagen, Denmark. *email: Juleen.Zierath@ki.se

Exercise is a crucial preventative and interventional medicine, helping to preclude and ameliorate metabolic diseases and secondary ageing[1,2]. Inactivity, including sitting time, is an important and independent contributor to metabolic diseases and overall mortality[3,4]. Inactivity and lack of physical activity are major elements within a milieu of contributory factors for the global rise in metabolic diseases[5,6]. Such diseases are reaching epidemic proportions, and exercise often remains the most affordable and efficient intervention to improve metabolic health in many populations.

Despite the profound benefit of exercise for the treatment and prevention of metabolic disease, knowledge of the mechanisms by which exercise improves metabolic health is insufficient. Differing exercise modalities such as resistance training, moderate intensity aerobic training, and high intensity interval training (HIIT) have overlapping, and discrete physiological outcomes, although the divergences in the signal transduction pathways are not fully understood[7,8]. Furthermore, the physiological response to exercise training varies between individuals[9–11], and exercise physiologists have scarce predictive tools at their disposal in this regard. A better understanding of the metabolic and cellular effects of exercise, coupled with advances in the characterization of the human genome, could lead to improved personalized/targeted exercise interventions. In addition, elucidation of the myriad of molecular changes induced by divergent exercise modalities may accelerate the discovery of pharmaceutical targets to improve metabolic health.

Attempting to answer these, and other biological questions, a rapid advancement in technology has allowed wide-scale use of omics. This has helped illuminate genomic regulation in response to external stimuli such as disease or exercise. However, analyses of these data remain nascent and implementation is not yet optimal. Over 60 studies have been published regarding skeletal muscle transcriptomic responses to different modes of exercise in various populations. The amount of biological data generated often surpasses the ability of researchers to comprehensively process it, leading to underutilized funding and endeavor. Instead, this public data remains a unique, unexploited resource.

Here, we propose a global approach to interrogate transcriptomic data using a meta-analysis of currently available exercise response datasets. This analysis is performed with the aim of providing a comprehensive resource for investigating the molecular and cellular effects of exercise, in addition to highlighting divergent transcriptomic responses of skeletal muscle to differing modalities of exercise, inactivity, and between phenotypically distinct individuals. This meta-analysis approach allows us to identify the involvement of nuclear receptor subfamily 4 group A member 3 (NR4A3) in the response to inactivity and characterize the effects of this gene on exercise-induced metabolic responses in skeletal muscle.

## Results

**Transcriptomic responses to aerobic and resistance exercise.** Publicly available datasets of the skeletal muscle transcriptomic response to resistance or aerobic exercise and inactivity in human volunteers were included in this meta-analysis (Supplementary Fig. 1). The meta-analysis (MetaMEx) included data from 12 studies of acute aerobic exercise, 8 studies of acute resistance exercise, 11 studies of aerobic-based exercise training and 13 studies of resistance-based exercise training. In addition, data from 6 studies assessing the effects of inactivity on the skeletal muscle transcriptome were included. In total, the meta-analysis represents data from more than 1100 individuals. Study characteristics are reported in Table 1 and the references to the studies are included in Table 2. All studies were annotated by skeletal

muscle type (*vastus lateralis*, *biceps brachii*, or *quadriceps femoris*) sex, age, fitness level and diseases state. Because of the limited number and heterogeneity of the studies, we distinguished between healthy individuals and those that are overweight/obese, type 2 diabetic or presenting with metabolic dysregulation such as dyslipidemia or impaired glucose tolerance. We labeled these two groups as Healthy and Metabolically Impaired, respectively. Studies were predominantly composed of male subjects of similar age, and the age span was smallest in the studies assessing acute aerobic exercise and inactivity (Table 1 and Supplementary Fig. 1). In the studies comparing metabolically impaired versus healthy individuals, body mass index (BMI) was higher, but the age range was similar. Overall, MetaMEx provides the most extensive dataset to date on the transcriptomic response of skeletal muscle to inactivity, acute exercise and exercise training.

**The MetaMEx database.** We validated the approach by interrogating MetaMEx for canonical exercise-induced genes. PGC1α (*PPARGC1A*) is a central regulator of skeletal muscle adaptation to exercise and is increased with acute exercise in rodents[12], although this is not a consistent finding across all human studies. Our meta-analysis confirmed that *PPARGC1A* is increased 2.3-fold (95% CI [1.6, 3.5]) after acute aerobic and 1.8-fold (95% CI [1.6, 2.2]) after acute resistance exercise (Fig. 1). *PPARGC1A* was consistently decreased 25% by inactivity. Exercise-induced changes in *PPARGC1A* expression was greatest (4.4-fold, 95% CI [3.0, 6.4]) in studies where skeletal muscle biopsies were taken after a recovery period (>2 h, REC) compared with immediately after exercise (<30 min, IMM). Moreover, *PPARGC1A* expression was modestly or not significantly altered after exercise training, suggesting that this gene is transiently induced in response to exercise. Our meta-analysis provides insight into the regulation of *PPARGC1A* mRNA and explains some of the discrepancies across studies.

To solve the problem of data accessibility, we have made MetaMEx available to the wider research community (www.metamex.eu), allowing users to interrogate the behavior and connectivity of specific genes across exercise studies. Any gene of interest can be tested in a similar fashion as *PPARGC1A* and the dataset is available for download. Thus, we provide a unique validation tool to meta-analyze changes in single genes across exercise and inactivity studies with various phenotypical data.

**Meta-analysis of skeletal muscle transcriptomic studies.** A principal component analysis (PCA) identified discrete clustering of gene responses based on intervention (Fig. 2a). Studies assessing the effects of acute aerobic and resistance exercise cluster together and away from studies assessing the effects of exercise training and inactivity.

Confirming the PCA, a chord plot revealed important overlap between acute aerobic and resistance studies, but few genes common between acute and training studies (Fig. 2b). A correlation matrix of the fold-change from all studies using all common genes (Fig. 2c) demonstrated correlations and clustering of acute studies with each other, including aerobic and resistance exercise. Similarly, most training protocols correlated with each other, irrespective of exercise modality. Overall, a clear segregation of the response to acute exercise, training and inactivity was observed, but no clear difference between resistance and aerobic exercise was noted.

We further used MetaMEx to perform a full meta-analysis of all transcripts. Restricted maximum likelihood was used to compute the fold-change and significance for each individual exercise- or inactivity-responsive gene. After adjustment for multiple comparisons, the number of significantly modified genes

**Table 1 Description of the study cohorts included in the meta-analysis.**

|  | Acute aerobic | Acute resistance | Inactivity | Training aerobic | Training resistance | Training aerobic | Training resistance |
|---|---|---|---|---|---|---|---|
| Group | HLY | HLY | HLY | HLY | HLY | MTI | MTI |
| Studies | 12 | 8 | 7 | 11 | 13 | 8 | 3 |
| Total females | 13 | 34 | 54 | 6 | 56 | 45 | 30 |
| Total males | 124 | 97 | 71 | 104 | 90 | 86 | 19 |
| Total undefined | 0 | 0 | 0 | 3 | 67 | 0 | 0 |
| % females | 9.5 | 26.0 | 43.2 | 5.5 | 38.4 | 34.4 | 61.2 |
| Age, mean ± sd | 30.7 ± 11.8 | 39.5 ± 27.0 | 30.6 ± 16.5 | 43.7 ± 19.7 | 46.7 ± 23.4 | 50.8 ± 8.9 | 56.2 ± 15.9 |
| BMI, mean ± sd | 24.9 ± 1.5 | 24.6 ± 0.7 | 23.2 ± 1.2 | 24.7 ± 1.4 | 25.8 ± 2 | 34 ± 6.2 | 33.8 ± 7 |

Study groups were composed of healthy (HLY) or metabolically impaired (MTI) individuals

was much higher than obtained in individual studies, demonstrating the power of the meta-analysis. Our analysis also demonstrated that each intervention modified the expression of select subsets of genes (Fig. 2d). We calculated the total number of responsive genes (FDR < 0.1%) for each perturbation and found 897 for acute aerobic, 2404 for acute resistance, 1576 for inactivity, 82 for aerobic training and 2049 for resistance training (Fig. 2e–i). We found acute aerobic and acute resistance exercise changed 360 genes in common, whereas aerobic training and resistance training changed 25 genes in common.

The meta-analysis identifies exercise- and inactivity-responsive genes and provides targets for future research by identifying many genes not previously studied in the context of exercise (Table 3). The top 5 up- and downregulated genes for each perturbation are presented in Fig. 3a. Changes in *NR4A3* expression have been described after acute aerobic exercise[12–14], although its specific role in orchestrating exercise-adaptations in skeletal muscle remains unclear. The transcriptional regulator *MAFF* (MAF BZIP Transcription Factor F) and the stress responsive protein *GADD45G* (Growth Arrest And DNA Damage Inducible Gamma) have yet to be studied in the context of exercise. For instance, *LMOD1* (Leiomodin 1) is commonly decreased after either aerobic or resistance training. Leiomodins are actin filament nucleators in muscle cells[15], but a role in skeletal muscle adaptations after exercise training is unknown. To further validate these results, we measured several of the top exercise-responsive genes (Table 3) in skeletal muscle from independent cohorts of healthy volunteers following aerobic and resistance exercise. The response of these modality-specific genes determined by qPCR analysis in the validation cohorts was highly correlated with MetaMEx (Supplementary Fig. 2). While this correlation was significant, many individual gene responses did not reach significance in the smaller validation cohorts, which was especially evident in response to training. Changes in mRNA expression after training are subtler than acute exercise and the statistical power given by the meta-analysis allows for the identification of many previously uncharacterized exercise-responsive genes.

Gene ontology was performed to characterize the pathways regulated by the different exercise protocols (Fig. 3b). Either acute aerobic or resistance exercise triggers an upregulation of apoptotic processes and kinase activity, but only acute aerobic exercise alters the expression of genes associated with vascular development. Inactivity was associated with a reduction in the expression of genes involved in mitochondrial processes and ATP production and increased expression of genes associated with ubiquitination. Conversely, aerobic training triggered an increase in mRNA of genes related to metabolic pathways and mitochondrial function, and resistance training was associated with increased mRNA of genes involved in extracellular matrix remodeling.

Exercise training improves metabolic flexibility by improving skeletal muscle insulin sensitivity, glucose and lipid metabolism, as well as mitochondrial function[2,16]. We, therefore, interrogated the database in a targeted manner for an effect of exercise on these processes. Aerobic exercise training, but not resistance training, upregulated the expression of genes coding for lipid regulating enzymes, while inactivity downregulated mRNA levels (Fig. 3c). Conversely, acute aerobic exercise and acute resistance exercise triggered an increase in mRNA of AMPK subunits, but other enzymes involved in lipid metabolism were unaltered. Mitochondrial complexes were increased by aerobic but not resistance training, while inactivity downregulated mRNA levels (Fig. 3d). Moreover, exercise training had a greater effect than acute exercise on the expression of genes coding for mitochondrial complexes.

Exercise triggers the remodeling of muscle tissue, with local inflammation playing a role to increase skeletal muscle mass and fiber composition[17]. The meta-analysis revealed that acute aerobic exercise elevated mRNA of many cytokines (Fig. 3e), in particular, the monocyte attractants *CCL2* (MCP-1) and *CXCL2* (Gro-β). Conversely, acute resistance exercise elevated *CCL2* mRNA, but did not increase the expression of other cytokines. Exercise training did not affect the mRNA level of these cytokines, presumably because the biopsies were generally collected 48 h after the last exercise bout, at a time where the inflammatory response had likely subsided. Myostatin (MSTN) is an inhibitor of muscle growth and differentiation[18]. We found that all of the exercise protocols reduced *MSTN* mRNA, whereas levels were increased with inactivity. We also found that all exercise protocols modulated myosin heavy and light chain mRNA (Fig. 3f). Most myosin chain isoforms were reduced after exercise training, with both aerobic and resistance exercise reducing *MYH1* and *MYH4* mRNA, and resistance training also reducing *MYH7*, *MYL2*, and *MYL3* mRNA. Conversely, *MYL6B* mRNA was increased by both aerobic and resistance exercise. Inactivity was associated with an inverse response of the myosin heavy chain profile compared with aerobic training.

**NR4A3 regulates skeletal muscle response to inactivity.** Amongst the hundreds of genes identified in MetaMEx, we focused on those modified by aerobic or resistance exercise and inactivity (Fig. 4a). We found that expression of *DNAJA4*, *KLHL40*, *NR4A3*, and *VGLL2* was increased by acute exercise and decreased by inactivity and further validated these genes in an independent cohort of acute aerobic exercise (Fig. 4b). Electrical pulse stimulation of primary human muscle cells in vitro mimics several of the exercise responses observed in adult skeletal muscle, such as increased gene expression and glucose uptake[19]. Of the four genes modified by exercise and inactivity, only *NR4A3* mRNA was increased after electrical pulse stimulation in human primary skeletal muscle cells (Fig. 4c). *NR4A3* responded to electrical pulse stimulation in an intensity and time-dependent manner (Fig. 5a). Electrical pulse stimulation increased glucose

**Table 2 Studies included in the MetaMEx database.**

| Protocol | GEO | Author | Reference | DOI |
|---|---|---|---|---|
| Acute aerobic | GSE4247 | Mahoney DJ, 2005 | [13] | https://doi.org/10.1096/fj.04-3149fje |
| Acute aerobic | GSE27285 | Rowlands DS, 2011 | [44] | https://doi.org/10.1152/physiolgenomics.00073.2011 |
| Acute aerobic | GSE33603 | Crane JD, 2012 | [45] | https://doi.org/10.1126/scitranslmed.3002882 |
| Acute aerobic | GSE41769 | Catoire M, 2012 | [46] | https://doi.org/10.1371/journal.pone.0051066 |
| Acute aerobic | GSE43219 | McLean CS, 2015 | [47] | https://doi.org/10.1371/journal.pone.0127089 |
| Acute aerobic | GSE43856 | Neubauer O, 2013 | [48] | https://doi.org/10.1152/japplphysiol.00143.2013 |
| Acute aerobic | GSE44818 | Rowlands DS, 2016 | [49] | https://doi.org/10.1152/physiolgenomics.00068.2015 |
| Acute aerobic | GSE59088 | Vissing K, 2014 | [50] | https://doi.org/10.1038/sdata.2014.41 |
| Acute aerobic | GSE59363 | Hansen JS, 2015 | [51] | https://doi.org/10.1007/s00125-015-3584-x |
| Acute aerobic | GSE68585 | Coletta DK, 2016 | [52] | https://doi.org/10.1371/journal.pone.0160327 |
| Acute aerobic | GSE71972 | Romero SA, 2016 | [53] | https://doi.org/10.1113/JP272177 |
| Acute aerobic | GSE86931 | Popov DV, 2015 | [54] | https://doi.org/10.1530/JME-15-0150 |
| Acute aerobic | GSE87748 | Pattamaprapanont P, 2016 | [14] | https://doi.org/10.3389/fendo.2016.00165 |
| Acute resistance | GSE1832 | Zambon AC, 2003 | [55] | https://doi.org/10.1186/gb-2003-4-10-r61 |
| Acute resistance | GSE4249 | Mohoney DJ, 2009 | [56] | https://doi.org/10.1152/ajpregu.00847.2007 |
| Acute resistance | GSE7286 | Kostek MC, 2007 | [57] | https://doi.org/10.1152/physiolgenomics.00151.2006 |
| Acute resistance | GSE19062 | MacNeil LG, 2010 | [58] | https://doi.org/10.1371/journal.pone.0010695 |
| Acute resistance | GSE23697 | Hyldahl RD, 2011 | [59] | https://doi.org/10.1096/fj.10-177105 |
| Acute resistance | GSE24235 | Liu D, 2010 | [60] | https://doi.org/10.1186/1471-2164-11-659 |
| Acute resistance | GSE28422 | Raue U, 2012 | [61] | https://doi.org/10.1152/japplphysiol.00435.2011 |
| Acute resistance | GSE59088 | Vissing K, 2014 | [50] | https://doi.org/10.1038/sdata.2014.41 |
| Inactivity | GSE14798 | Chopard A, 2009 | [62] | https://doi.org/10.1152/physiolgenomics.00036.2009 |
| Inactivity | GSE14901 | Abadi A, 2009 | [63] | https://doi.org/10.1371/journal.pone.0006518 |
| Inactivity | GSE21496 | Reich KA, 2010 | [64] | https://doi.org/10.1152/japplphysiol.00444.2010 |
| Inactivity | GSE24215 | Alibegovic AC, 2010 | [65] | https://doi.org/10.1152/ajpendo.00590.2009 |
| Inactivity | GSE33886 | Lammers G, 2012 | [66] | https://doi.org/10.1113/expphysiol.2012.068726 |
| Inactivity | GSE104999 | Rullman E, 2016 | [67] | https://doi.org/10.14814/phy2.12753 |
| Inactivity | GSE113165 | Mahmassani ZS, 2019 | [68] | https://doi.org/10.1152/japplphysiol.01093.2018 |
| Training aerobic | GSE1295 | Hittel DS, 2005 | [69] | https://doi.org/10.1152/japplphysiol.00331.2004 |
| Training aerobic | GSE1718 | Teran-Garcia M, 2005 | [70] | https://doi.org/10.1152/ajpendo.00467.2004 |
| Training aerobic | GSE1786 | Radom-Aizik S, 2005 | [71] | https://doi.org/10.1249/01.mss.0000181838.96815.4d |
| Training aerobic | GSE9103 | Lanza IR, 2008 | [72] | https://doi.org/10.2337/db08-0349 |
| Training aerobic | GSE9405 | Stepto NK, 2009 | [73] | https://doi.org/10.1249/MSS.0b013e31818c6be9 |
| Training aerobic | GSE20319 | Leskinen T, 2010 | [74] | https://doi.org/10.1371/journal.pone.0012609 |
| Training aerobic | GSE24215 | Alibegovic AC, 2010 | [65] | https://doi.org/10.1152/ajpendo.00590.2009 |
| Training aerobic | GSE27536 | Turan N, 2011 | [75] | https://doi.org/10.1371/journal.pcbi.1002129 |
| Training aerobic | GSE27543 | | | |
| Training aerobic | GSE35661 | Keller P, 2011 | [76] | https://doi.org/10.1152/japplphysiol.00634.2010 |
| Training aerobic | GSE40551 | Engeli S, 2012 | [77] | https://doi.org/10.1172/JCI64526 |
| Training aerobic | GSE43760 | Verheggen RJHM, 2016 | [78] | https://doi.org/10.1249/MSS.0000000000001035 |
| Training aerobic | GSE48278 | Huffman KM, 2014 | [79] | https://doi.org/10.1007/s00125-014-3343-4 |
| Training aerobic | GSE58249 | Sukala WR, 2012 | [80] | https://doi.org/10.1007/s00421-011-1978-0 |
| Training aerobic | GSE72462 | Böhm A, 2016 | [81] | https://doi.org/10.2337/db15-1723 |
| Training aerobic | GSE111551 | | | |
| Training combined | GSE14798 | Chopard A, 2009 | [62] | https://doi.org/10.1152/physiolgenomics.00036.2009 |
| Training combined | GSE19420 | van Tienen FHJ, 2012 | [82] | https://doi.org/10.1210/jc.2011-3454 |
| Training combined | GSE53598 | Catoire M, 2014 | [83] | https://doi.org/10.1152/physiolgenomics.00174.2013 |
| Training combined | GSE83352 | Barberio MD, 2016 | [84] | https://doi.org/10.1249/MSS.0000000000001041 |
| Training combined | GSE97084 | Robinson MM, 2017 | [8] | https://doi.org/10.1016/j.cmet.2017.02.009 |
| Training HIIT | GSE97084 | Robinson MM, 2017 | [8] | https://doi.org/10.1016/j.cmet.2017.02.009 |
| Training HIIT | GSE109657 | Miyamoto-Mikami E, 2018 | [85] | https://doi.org/10.1038/s41598-018-35115-x |
| Training resistance | GSE8479 | Melov S, 2007 | [86] | https://doi.org/10.1371/journal.pone.0000465 |
| Training resistance | GSE9405 | Stepto NK, 2009 | [73] | https://doi.org/10.1249/MSS.0b013e31818c6be9 |
| Training resistance | GSE16907 | Pöllänen E, 2010 | [87] | https://doi.org/10.1007/s11357-010-9140-1 |
| Training resistance | GSE24235 | Liu D, 2010 | [60] | https://doi.org/10.1186/1471-2164-11-659 |
| Training resistance | GSE28422 | Raue U, 2012 | [61] | https://doi.org/10.1152/japplphysiol.00435.2011 |
| Training resistance | GSE28998 | Gordon PM, 2012 | [88] | https://doi.org/10.1152/japplphysiol.00860.2011 |
| Training resistance | GSE45426 | Murton AJ, 2013 | [89] | https://doi.org/10.1152/japplphysiol.00426.2013 |
| Training resistance | GSE47881 | Phillips BE, 2013 | [90] | https://doi.org/10.1371/journal.pgen.1003389 |
| Training resistance | GSE48278 | Huffman KM, 2014 | [79] | https://doi.org/10.1007/s00125-014-3343-4 |
| Training resistance | GSE58249 | Sukala WR, 2012 | [80] | https://doi.org/10.1007/s00421-011-1978-0 |
| Training resistance | GSE97084 | Robinson MM, 2017 | [8] | https://doi.org/10.1016/j.cmet.2017.02.009 |
| Training resistance | GSE99963 | Laker RC, 2017 | [91] | https://doi.org/10.1038/s41598-017-15420-7 |
| Training resistance | GSE106865 | Damas F, 2018 | [92] | https://doi.org/10.1007/s00421-018-3984-y |
| Training resistance | GSE117525 | Hangelbroek RW, 2018 | [93] | https://doi.org/10.1002/jcsm.12099 |
| Training resistance | EMEXP740 | Tarnopolsky M, 2007 | [94] | https://doi.org/10.1093/gerona/62.10.1088 |

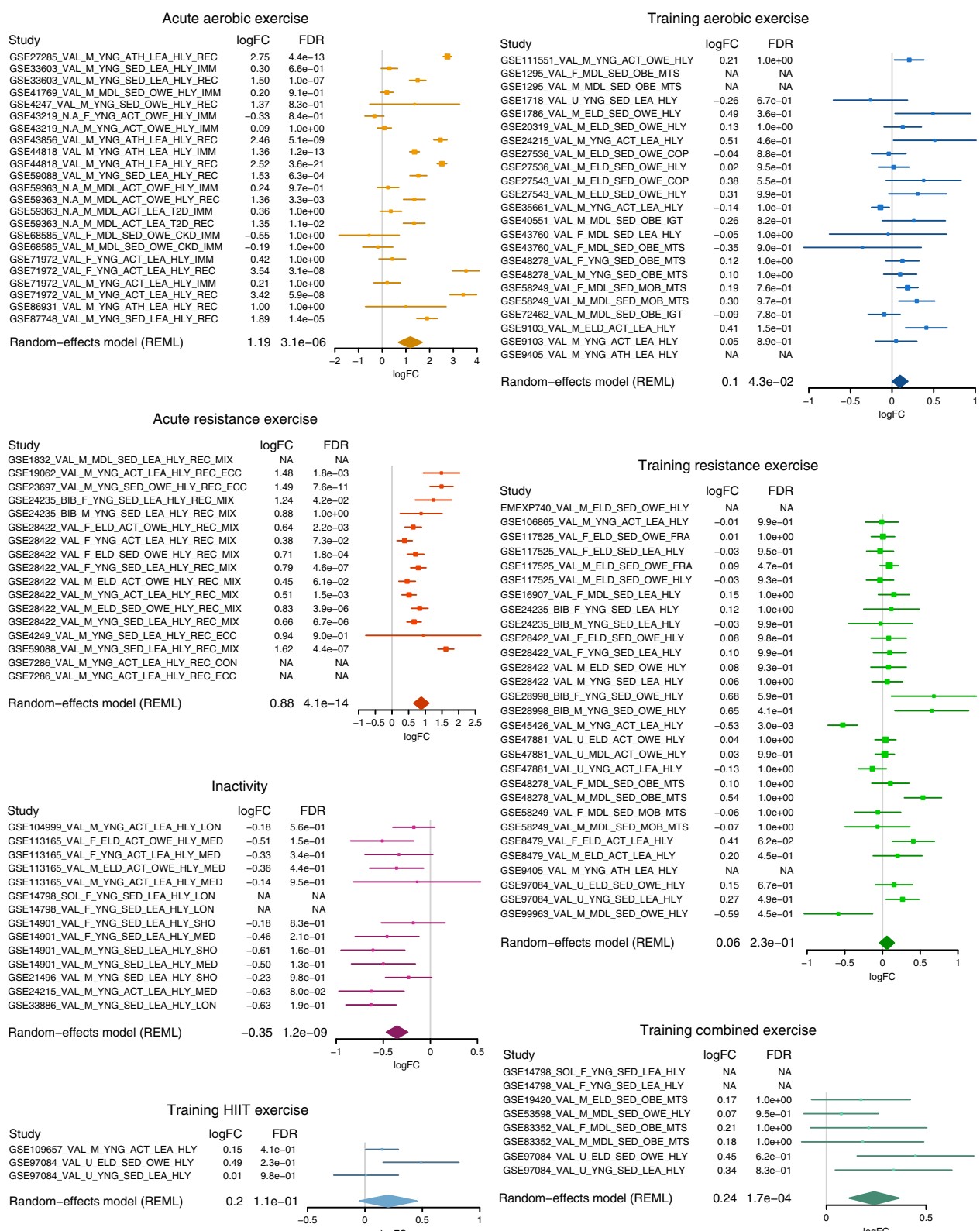

**Fig. 1 MetaMEx reveals the behavior of *PPARGC1A* across 66 transcriptomic studies.** The online tool MetaMEx (www.metamex.eu) allows for the quick interrogation of all published exercise and inactivity studies for a single gene. The analysis provides annotations of each study with respect to skeletal muscle type obtained, sex, age, fitness, weight, and metabolic status of the participants studied. The forest plot of individual statistics (fold-change, FDR, 95% confidence intervals), as well as the meta-analysis score is provided. In the case of HIIT training and combined exercise training protocols, the number of studies is insufficient to calculate meaningful meta-analysis statistics. NA: not available.

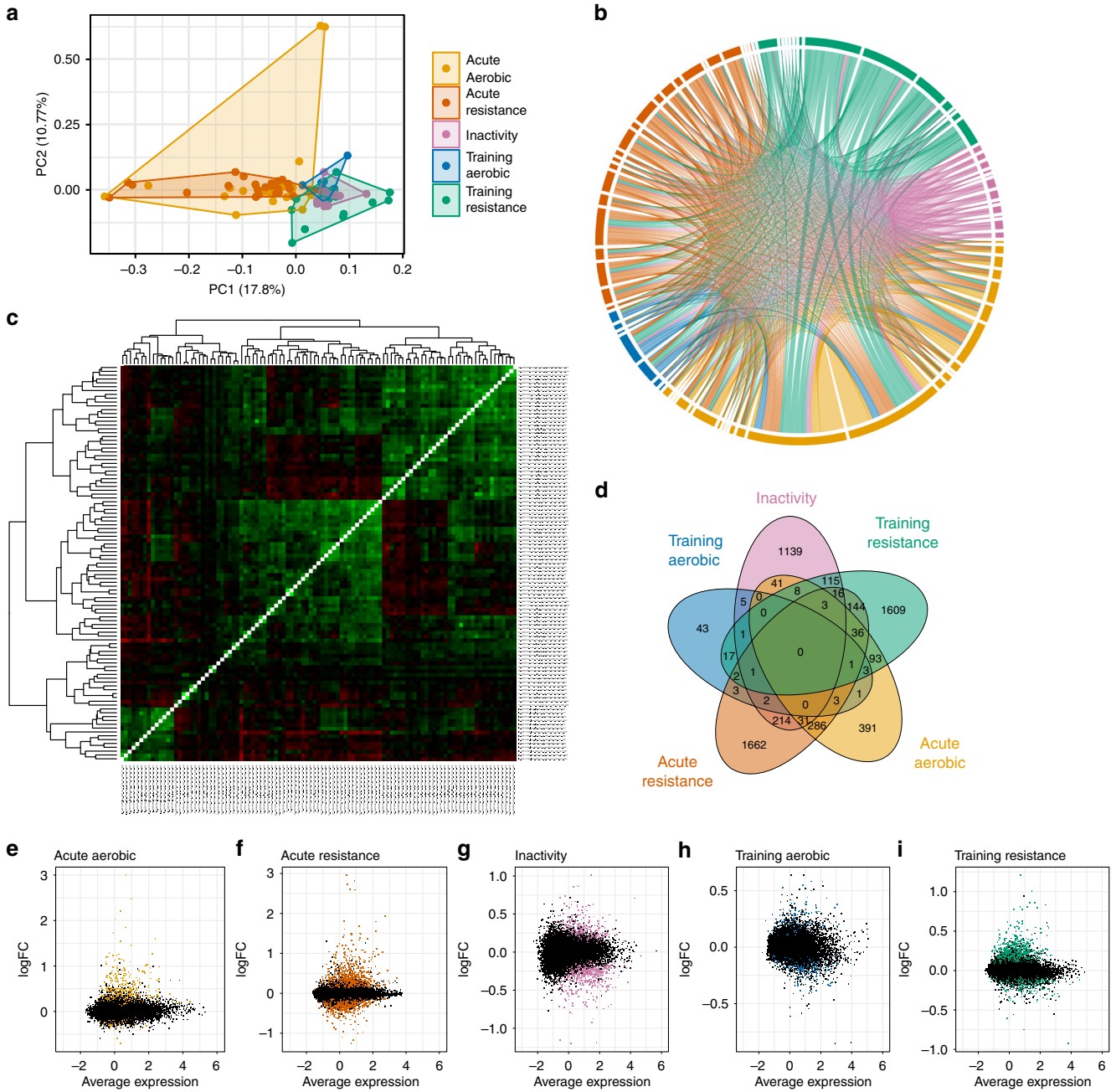

**Fig. 2 Inter-array comparisons separate acute exercise from training and inactivity.** All datasets of healthy individuals were compared with each other using a principle component analysis (**a**), a chord plot (**b**) and a correlation matrix of fold-changes (**c**). A Venn Diagram presents the overlap of the significantly (FDR < 1%) expressed genes (**d**). All genes are presented in M-plots (**e**–**i**) with significantly changed genes (FDR < 1%) represented with colored dots.

uptake in primary myotubes and silencing of *NR4A3* ablated this effect (Fig. 5b, c). Conversely, silencing of either *DNAJA4*, *KLHL40*, or *VGLL2* did not alter glucose uptake (Supplementary Fig. 3A, B).

In primary human skeletal muscle cells, *NR4A3* silencing altered exercise- and inactivity-responsive genes (Fig. 5d). The gene expression profile following *NR4A3* silencing was correlated with the transcriptomic response to inactivity as observed in MetaMEx (Fig. 5e). Silencing of either *DNAJA4*, *KLHL40*, or *VGLL2* also altered the expression of several genes, but not in a manner that correlated with the response to exercise or inactivity (Supplementary Fig. 3C–E). Consequently, we focused attention on *NR4A3* and validated a role of this gene in mitochondrial

function. Silencing *NR4A3* decreased basal and maximal oxygen consumption rate (OCR, Fig. 5f and Supplementary Fig. 4), shifting myotube metabolism towards a more quiescent phenotype (Fig. 5g). The decrease in oxygen consumption was associated with a decrease in mitochondrial oxidative phosphorylation complexes (Fig. 5h). These results establish a role for *NR4A3* in regulating the metabolic response to exercise. The beta-adrenergic receptor agonist salbutamol has been used as an exercise mimetic and upregulates NR4A3 mRNA[20]. Salbutamol increased glycolysis in myotubes in a *NR4A3*-dependent manner (Fig. 5i). Collectively, our results provide evidence that *NR4A3* regulates exercise/inactivity-responsive genes, and the metabolic response to contraction.

**Table 3 Top exercise and inactivity-responsive genes in healthy individuals.**

| | Acute aerobic, logFC | Acute aerobic, FDR | Acute resistance, logFC | Acute resistance, FDR | Inactivity, logFC | Inactivity, FDR | Training aerobic, logFC | Training aerobic, FDR | Training resistance, logFC | Training resistance, FDR |
|---|---|---|---|---|---|---|---|---|---|---|
| NR4A3 | 2.99 | 2.0E−07 | 2.95 | 8.2E−15 | −0.32 | 9.4E−02 | 0.13 | 5.7E−01 | −0.03 | 8.7E−01 |
| EGR1 | 2.47 | 3.8E−08 | 1.49 | 2.0E−04 | 0.42 | 9.7E−02 | 0.17 | 4.6E−01 | −0.04 | 8.8E−01 |
| FOS | 2.20 | 5.7E−06 | 1.41 | 1.5E−04 | 0.09 | 8.5E−01 | 0.07 | 8.9E−01 | −0.56 | 1.4E−01 |
| MAFF | 2.03 | 3.0E−05 | 2.60 | 1.7E−12 | 0.15 | 4.6E−01 | 0.14 | 2.1E−01 | 0.14 | 3.7E−01 |
| CYR61 | 1.84 | 9.0E−05 | 1.50 | 5.4E−06 | 0.30 | 3.6E−02 | 0.21 | 2.8E−01 | −0.25 | 4.0E−01 |
| GADD45G | −0.70 | 2.7E−03 | −1.25 | 2.3E−08 | 0.08 | 7.3E−01 | −0.08 | 8.1E−01 | −0.14 | 3.6E−01 |
| CA4 | −0.59 | 4.3E−03 | −0.23 | 2.3E−03 | 0.06 | 5.9E−01 | 0.03 | 9.2E−01 | 0.02 | 8.6E−01 |
| MSTN | −0.42 | 6.3E−03 | −0.96 | 5.0E−12 | 0.67 | 2.4E−09 | −0.32 | 1.4E−01 | −0.23 | 6.5E−02 |
| ADH1C | −0.39 | 2.4E−05 | −0.47 | 9.9E−04 | 0.24 | 7.7E−02 | −0.26 | 9.3E−02 | −0.11 | 2.6E−01 |
| LGALSL | −0.39 | 2.7E−04 | −0.07 | 5.9E−01 | −0.23 | 7.1E−03 | 0.02 | 9.1E−01 | 0.00 | 1.0E+00 |
| ANKRD1 | 0.94 | 2.3E−02 | 2.81 | 1.0E−11 | 0.41 | 3.8E−01 | 0.13 | 7.7E−01 | 0.14 | 6.2E−01 |
| ATF3 | 1.13 | 2.2E−04 | 2.73 | 7.7E−10 | −0.13 | 4.9E−01 | 0.18 | 2.8E−01 | 0.09 | 5.1E−01 |
| TNFRSF12A | 0.90 | 3.4E−04 | 2.60 | 3.8E−25 | −0.21 | 4.5E−01 | 0.23 | 2.0E−01 | −0.03 | 8.8E−01 |
| ARRDC2 | −0.33 | 7.5E−02 | −1.18 | 8.2E−13 | 0.63 | 1.1E−11 | −0.20 | 2.6E−01 | −0.13 | 1.7E−01 |
| LRRC66 | −0.13 | 2.8E−01 | −1.10 | 3.0E−07 | 0.15 | 3.8E−02 | −0.06 | 6.1E−01 | −0.16 | 2.8E−02 |
| CIART | −0.16 | 4.6E−01 | −1.00 | 4.0E−15 | −0.02 | 9.1E−01 | 0.19 | 4.5E−01 | −0.13 | 2.5E−01 |
| DDIT4 | 0.10 | 8.2E−01 | −0.96 | 4.9E−17 | 0.34 | 1.8E−02 | −0.22 | 4.3E−01 | −0.28 | 5.1E−02 |
| PFKFB3 | 0.41 | 1.2E−01 | −0.68 | 9.3E−05 | 1.01 | 1.3E−05 | −0.29 | 4.3E−01 | −0.17 | 6.0E−01 |
| CHRNA1 | 0.15 | 4.5E−01 | 0.12 | 6.2E−01 | 0.88 | 1.0E−08 | 0.19 | 4.7E−01 | 0.22 | 1.2E−02 |
| GADD45A | 0.89 | 8.4E−06 | 0.82 | 1.8E−03 | 0.86 | 1.8E−11 | 0.17 | 5.1E−01 | 0.16 | 5.2E−01 |
| CHRND | 0.13 | 3.6E−01 | 0.00 | 9.9E−01 | 0.85 | 4.8E−04 | −0.01 | 9.8E−01 | 0.14 | 1.5E−01 |
| HSD52 | | | | | 0.73 | 8.4E−04 | | | −0.42 | 1.9E−01 |
| CA14 | 0.01 | 9.8E−01 | −0.17 | 3.8E−01 | −1.20 | 9.7E−05 | 0.28 | 2.3E−01 | 0.24 | 2.2E−01 |
| NMRK2 | 0.36 | 1.1E−01 | 0.53 | 4.5E−02 | −1.19 | 5.4E−06 | 0.54 | 1.6E−01 | 0.83 | 1.8E−05 |
| SMCO1 | −0.09 | 3.7E−01 | 0.15 | 1.1E−01 | −0.87 | 5.2E−15 | −0.15 | 2.1E−01 | −0.35 | 8.2E−04 |
| COLQ | 0.16 | 3.1E−01 | 0.74 | 5.0E−06 | −0.81 | 4.8E−12 | 0.12 | 4.2E−01 | −0.15 | 9.3E−02 |
| EXTL1 | −0.02 | 9.0E−01 | −0.15 | 5.2E−02 | −0.79 | 5.1E−08 | 0.21 | 7.4E−02 | −0.10 | 1.9E−01 |
| CDK2 | 0.19 | 2.9E−03 | 0.09 | 4.0E−01 | −0.06 | 3.9E−01 | 0.40 | 1.4E−05 | 0.21 | 3.5E−05 |
| ITGA6 | 0.06 | 6.0E−01 | 0.02 | 9.2E−01 | 0.08 | 4.6E−01 | 0.34 | 5.1E−04 | 0.17 | 9.5E−04 |
| SCO1 | 0.04 | 5.9E−01 | −0.08 | 1.5E−01 | 0.06 | 3.7E−01 | 0.32 | 7.1E−03 | 0.02 | 8.0E−01 |
| SRGN | 0.45 | 3.8E−08 | 0.05 | 6.4E−01 | −0.06 | 7.1E−01 | 0.32 | 5.1E−03 | 0.25 | 6.6E−03 |
| MYH9 | 0.61 | 1.6E−03 | 0.62 | 2.0E−08 | −0.06 | 7.7E−01 | 0.31 | 3.3E−03 | 0.24 | 1.8E−02 |
| HIST1H1C | −0.20 | 2.4E−01 | 0.00 | 1.0E+00 | 0.26 | 2.5E−06 | −0.40 | 2.6E−03 | −0.11 | 1.1E−01 |
| SH3RF2 | −0.14 | 3.7E−01 | 0.03 | 8.4E−01 | 0.55 | 1.1E−06 | −0.34 | 7.3E−03 | −0.27 | 8.7E−03 |
| FHL3 | 0.01 | 9.5E−01 | −0.01 | 9.1E−01 | −0.02 | 9.4E−01 | −0.32 | 6.1E−01 | −0.06 | 7.0E−01 |
| SCNIB | 0.08 | 3.2E−01 | −0.07 | 2.6E−01 | −0.01 | 9.3E−01 | −0.30 | 4.6E−03 | 0.19 | 9.2E−03 |
| DYRK2 | −0.30 | 6.0E−03 | 0.11 | 6.5E−01 | 0.04 | 8.3E−01 | −0.28 | 8.8E−03 | −0.09 | 7.8E−02 |
| MXRA5 | 0.11 | 6.7E−01 | −0.15 | 6.5E−01 | −0.27 | 3.3E−01 | 0.63 | 2.9E−01 | 1.20 | 8.5E−06 |
| COL3A1 | 0.17 | 4.6E−01 | 0.10 | 7.2E−01 | −0.05 | 9.4E−01 | 0.51 | 2.0E−01 | 1.01 | 1.7E−11 |
| COL4A1 | 0.38 | 1.3E−02 | 0.24 | 3.3E−01 | −0.10 | 5.3E−01 | 0.63 | 5.9E−02 | 0.86 | 1.3E−09 |
| COL1A2 | 0.17 | 3.4E−01 | 0.08 | 7.4E−01 | 0.12 | 7.7E−01 | 0.38 | 2.2E−01 | 0.86 | 2.5E−07 |
| MYH1 | −0.07 | 8.2E−01 | −0.32 | 1.3E−01 | 0.70 | 1.8E−03 | −0.84 | 1.6E−01 | −0.92 | 8.3E−04 |
| CALML6 | −0.29 | 2.5E−01 | −0.06 | 8.3E−01 | 0.23 | 3.7E−01 | −0.59 | 1.5E−01 | −0.75 | 1.3E−05 |
| METTL21C | −0.01 | 9.5E−01 | −0.30 | 3.4E−01 | 0.11 | 6.8E−01 | −0.17 | 1.5E−01 | −0.60 | 7.9E−03 |
| MYLK2 | −0.12 | 3.0E−01 | −0.23 | 9.7E−03 | 0.09 | 6.6E−01 | −0.35 | 4.5E−02 | −0.45 | 2.9E−06 |
| PAIP2B | −0.15 | 9.1E−02 | −0.16 | 1.5E−01 | −0.34 | 4.2E−03 | −0.01 | 9.8E−01 | −0.40 | 2.2E−04 |

Table presents the log(fold-change) before-after intervention and the false discovery rate (FDR)

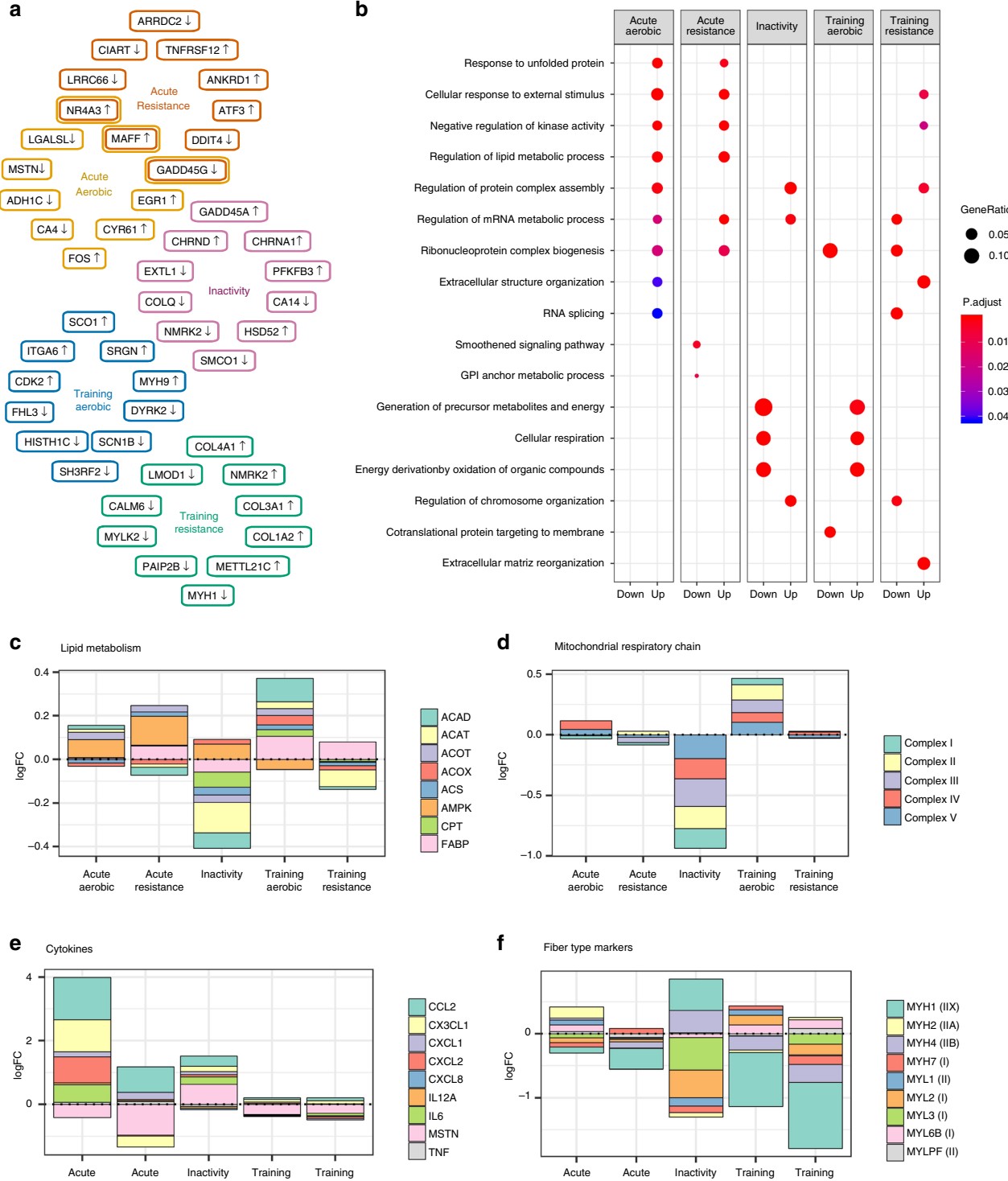

**Fig. 3 Genes and pathways altered by exercise and inactivity.** The top 5 up- and downregulated genes in each protocol and their overlap was calculated (**a**) and gene ontology analysis was calculated based on genes with FDR < 0.01 (**b**). Genes corresponding to the proteins of interest were collected from the KEGG database and fold-changes were added to present the overall modification of enzymes involved in pathways. The behavior of lipid metabolism signaling (**c**), mitochondrial respiration (**d**), inflammation (**e**), and muscle fiber composition (**f**) are presented.

**Exercise response in metabolically impaired individuals.** In addition to defining the skeletal muscle transcriptomic response to different modes of exercise, MetaMEx can also be used to compare metabolically impaired (obese and/or type 2 diabetes) individuals with healthy volunteers. Due to the limited number of studies of metabolically impaired individuals, a meta-analysis could only be performed on the training studies. We selected

aerobic and resistance training studies such that the healthy and metabolically impaired groups were matched for age and excluded studies with obvious differences in either training protocols or skeletal muscle group sampled (Supplementary Table 2). Our analysis included six studies of aerobic training and two studies of resistance training comparing obese and/or type 2 diabetic versus healthy volunteers. The PCA separated aerobic and resistance

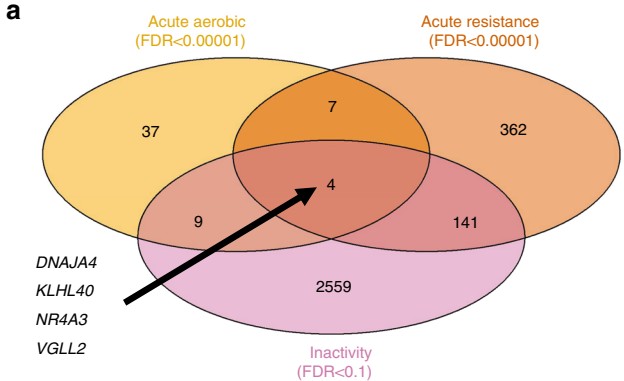

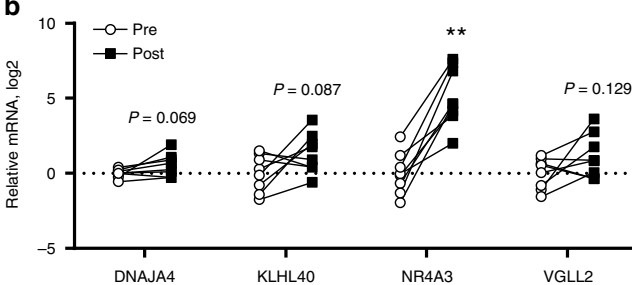

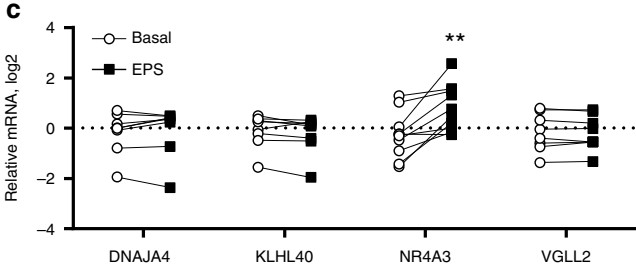

**Fig. 4 *DNAJA4*, *KLHL40*, *NR4A3*, and *VGLL2* respond to exercise and inactivity. a** Genes significantly modified by acute aerobic and resistance exercise and inactivity were overlapped in a Venn Diagram. **b** *DNAJA4*, *KLHL40*, *NR4A3*, and *VGLL2* were validated in an independent cohort of pre- and post-acute aerobic exercise. Individual paired *t*-tests vs pre, $n = 8$ biologically independent volunteers, **$p < 0.01$. **c** *DNAJA4*, *NR4A3*, *KLHL40*, and *VGLL2* gene expression following electrical pulse stimulation in primary human myotubes. Individual paired *t*-tests vs basal, $n = 8$ biologically independent primary cells from different donors, **$p < 0.01$.

training in healthy individuals, but not in metabolically impaired individuals (Fig. 6a). The meta-analysis identified hundreds of differentially regulated genes, from which the top 250 genes ($p < 0.01$) were selected based on absolute fold-change. In healthy individuals, aerobic and resistance training induced distinct gene profiles, differing between the metabolically impaired and healthy individuals (Fig. 6b). Pathway analysis revealed that exercise triggered differential responses such as aerobic training-induced matrix remodeling, a feature only observed in resistance training in healthy individuals (Fig. 6c). The response of classical exercise-responsive pathways such as mitochondrial complexes, as well as lipid and glucose oxidation (Supplementary Fig. 5), was similar between healthy and metabolically impaired individuals. However, we also identified a subset of genes specifically altered in healthy volunteers that were not changed in metabolically impaired individuals (Table 4). We identified genes involved in transcriptional regulation (*KANSL3*, *ARNT*, *TOP2B*), inflammation (*IGIP*, *PTGDS*) and lipid transport (*ABCG1*) were altered in

metabolically impaired individuals after exercise. These genes have not been studied in the context of exercise, and the specificity to obese/type 2 diabetic individuals may provide targets to optimize exercise in the metabolically impaired population.

## Discussion

Exercise is a vital tool in the fight against the global epidemic of metabolic disease. Our study presents a unique resource with the aim of uniting valuable, publicly available data in order to drive new hypothesis and innovative discoveries. The statistical power of this analysis has allowed identification of several previously unrecognized or understudied pathways, including divergent transcriptomic responses to exercise modalities, inactivity, and between phenotypically distinct individuals. This analysis identified NR4A3 as a specific, key target in response to inactivity and acute exercise that drives some of the metabolic changes associated with these perturbations. MetaMEx provides a central repository for future investigations, aiming to enhance the coordination of exercise transcriptomics. These data, and the putative legacy of MetaMEx, will help to address some of the key questions remaining within exercise physiology research, including identifying modality-dependent pathways and characterizing discrete or pathological responses to exercise in various populations.

Improved technology has permitted a rapid expansion of the capacity to examine the genomic response to exercise[21]. While emerging investigations have produced several important discoveries, progression in the field may be accelerated by coordinated research efforts. Many clinical studies designed to assess the effects of exercise on functional and molecular outcomes include small cohorts and may be underpowered to detect subtle, but biologically relevant changes. This is precisely what we observe in our validation cohorts, where most of the genes identified in MetaMEx did not reach significance, although similar corelative changes were detected. Most exercise studies include homogeneous populations, with a recruitment bias towards male volunteers, and a narrow age range, within either young, middle-aged or elderly populations (Table 1). While homogenous cohorts have the advantage of increased statistical power, they come with the risk of artefactual observations. Our meta-analysis approach surmounts these limitations by pooling populations of all ages, sex, BMI and metabolic parameters within a robust cohort of healthy volunteers. This eliminates single-study artefacts, and maps the global response of skeletal muscle to different modes of exercise and inactivity.

Meta-analyses are commonly used for clinical studies, and have gained popularity to improve statistical power, particularly for genetic and transcriptomic studies[22]. Here, we present a validated online tool, MetaMEx www.metamex.eu, with the advantage that additional studies can be incorporated, so that statistical power will increase as researchers publish additional datasets. This approach surpasses the analyses of single arrays, and thereby increases the statistical power to uncover new biology involved in the plasticity of skeletal muscle by allowing for the characterization of the transcriptomic fingerprint in response to intervention modalities in phenotypically distinct cohorts. This work provides an open access database of exercise transcriptomic studies curated and fully annotated by age, sex, BMI, and metabolic disease status. The benefits of this centralized database include the ability to identify knowledge gaps and conduct initial hypothesis testing, leading to improved design of prospective intervention studies.

Inactivity, type 2 diabetes and obesity are associated with insulin resistance, while regular physical exercise improves glucose uptake and insulin sensitivity[2]. The mechanisms underlying the deleterious effects of physical inactivity differ from those

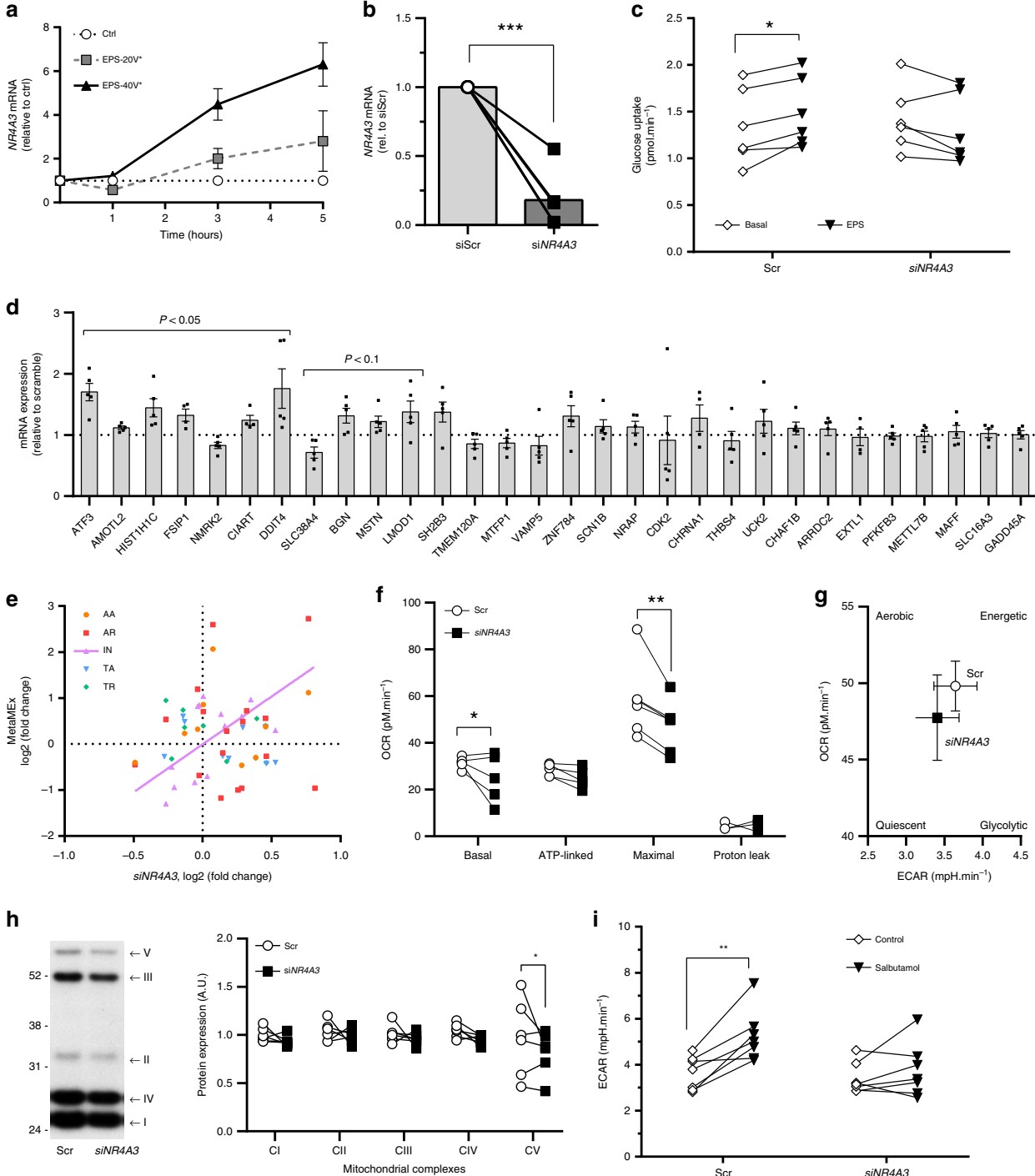

**Fig. 5 NR4A3 regulates the metabolic response to in vitro exercise. a** NR4A3 responds in a time- and intensity-dependent manner to electrical pulse stimulation. Data are mean ± SEM, $n = 3$ biologically independent primary cells from different donors, two-way ANOVA (time, intensity), *overall effect $p <$ 0.05. **b** Silencing efficiency using siRNA against NR4A3, $n = 4$ biologically independent primary cells from different donors, individual paired $t$-test vs scramble, ***$p < 0.001$. **c** Electrical pulse stimulation-induced glucose uptake is abolished after NR4A3 silencing. Two-way ANOVA (si*NR4A3*, EPS), $n = 6$ biologically independent primary cells from different donors, *$p < 0.05$. **d** Silencing *NR4A3* using siRNA modifies the mRNA levels of exercise- and inactivity-responsive genes. Data are mean ± SEM, $n = 4$ biologically independent primary cells from different donors, individual paired $t$-test vs scramble. **e** Silencing of *NR4A3* correlates with inactivity observed in MetaMEx. **f** Reduction of *NR4A3* level impairs basal and maximal oxygen consumption measured by Seahorse XF analysis. Individual paired $t$-tests vs scramble, $n = 5$ biologically independent primary cells from different donors, *$p < 0.05$, **$p < 0.01$. **g** Silencing of *NR4A3* leads to a drift of muscle cells towards a more quiescent phenotype. Data are mean ± SEM. **h** Silencing of *NR4A3* decreases the abundance of mitochondrial complexes. Representative blot and quantification, $n = 6$ biologically independent primary cells from different donors, two-way ANOVA, significant effect of silencing ($p = 0.018$), *$p < 0.05$ uncorrected Fisher's LSD post-test. **i** The increase in glycolysis (ECAR) induced by beta-adrenergic stimulation (20 μM salbutamol for 3 h) is impaired in the absence of *NR4A3*. SeaHorse XF experiment, two-way ANOVA (si*NR4A3*, Salbutamol), $n = 7$ biologically independent primary cells from different donors, **$p < 0.01$. AA: acute aerobic, AR: acute resistance, IN: inactivity, TA: training aerobic, TR: training resistance.

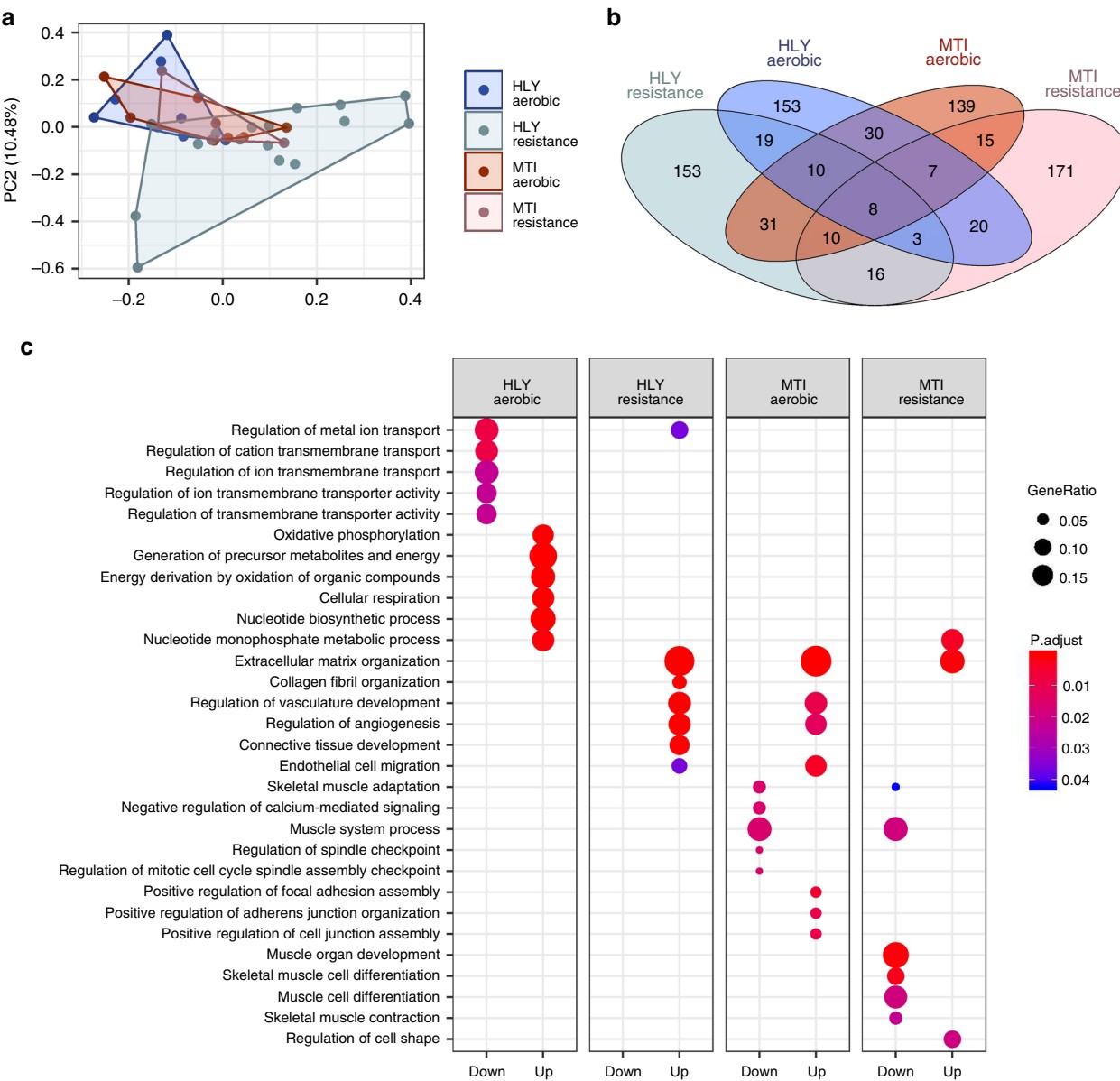

**Fig. 6 Differential response to exercise training in metabolically impaired individuals.** To compare healthy (HLY) to metabolically impaired (MTI) individuals, a principle component analysis was performed (**a**) and the significantly regulated genes (FDR < 0.1) overlapped in a Venn diagram (**b**). Gene ontology analysis calculated based on genes with FDR < 0.1 demonstrated a differential response of metabolically impaired individual to both aerobic and resistance training protocols (**c**).

conferring the benefits of physical activity[23]. We observed little overlap between individual genes regulated by aerobic or resistance training and genes modified by inactivity. However, gene ontology analysis revealed that pathways related to oxidative phosphorylation are decreased with inactivity and increased after aerobic training, suggesting that although different sets of genes are recruited in response to these divergent perturbations, they converge on mitochondrial function.

Exercise training is associated with profound changes in skeletal muscle, including increased abundance of glucose transporters, activation of AMPK and mitochondrial biogenesis[24]. MetaMEx allows for the validation of previous findings, such as the increase in *PPARGC1A* mRNA after acute exercise[25], as well as the increase in mitochondrial genes after aerobic training[24]. Our analysis also revealed insights into molecular exercise physiology. We found that *NR4A3* is the most acute aerobic and resistance exercise-responsive gene and is inversely regulated by

inactivity. Using cultured myotubes, we found that contraction directly upregulates *NR4A3* while silencing of this gene recapitulates the inactivity signature in MetaMEx and abolishes contraction-mediated glucose uptake. *NR4A3* silencing was associated with cellular-physiological outcomes that match those of inactivity such as reduced mitochondrial complex expression and decreased oxidative capacity[26,27]. In mice, skeletal muscle-specific overexpression of NR4A3 induces an oxidative, high-endurance phenotype with increased mitochondrial content and mtDNA copy number, elevated myoglobin, enhanced ATP production, and *PPARGC1A* gene expression[28]. This phenotype also includes morphological changes, with an increase in type II fatigue-resistant, oxidative fibers[29]. The mechanistic link between *NR4A3* and exercise-mediated metabolic responses may involve calcium ion signaling[30,31]. Our findings identify NR4A3 as a central regulator of the acute aerobic and resistance exercise response and the deleterious effects of inactivity.

**Table 4 Genes non-responsive to exercise in metabolically impaired individuals.**

| | Aerobic training healthy, logFC | Aerobic training healthy, FDR | Aerobic training obese/T2D, logFC | Aerobic training obese/T2D, FDR | Resistance training healthy, logFC | Resistance training healthy, FDR | Resistance training obese/T2D, logFC | Resistance training obese/T2D, FDR |
|---|---|---|---|---|---|---|---|---|
| GUCY1B1 | 0.26 | 7.8E − 03[a] | 0.09 | 9.7E − 01 | 0.11 | 2.0E − 01 | 0.02 | 1.0E + 00 |
| KANSL3 | − 0.09 | 7.8E − 03[a] | − 0.04 | 1.0E + 00 | 0.03 | 6.9E − 01 | 0.01 | 1.0E + 00 |
| ARNT | − 0.10 | 7.8E − 03[a] | − 0.02 | 1.0E + 00 | 0.01 | 9.3E − 01 | − 0.05 | 1.0E + 00 |
| TOP2B | − 0.20 | 7.1E − 03[a] | − 0.04 | 1.0E + 00 | − 0.09 | 5.5E − 02 | 0.00 | 1.0E + 00 |
| IGIP | − 0.19 | 7.1E − 03[a] | − 0.03 | 1.0E + 00 | − 0.10 | 1.6E − 01 | − 0.03 | 1.0E + 00 |
| COL6A6 | 0.24 | 7.9E − 02 | 0.04 | 8.1E − 01 | 0.36 | 6.2E − 04[a] | − 0.03 | 1.0E + 00 |
| APLNR | 0.22 | 3.9E − 01 | 0.18 | 7.8E − 01 | 0.36 | 6.5E − 03[a] | − 0.01 | 1.0E + 00 |
| COL4A2 | 0.53 | 2.8E − 01 | 0.33 | 1.3E − 01 | 0.53 | 9.7E − 04[a] | 0.19 | 1.0E + 00 |
| PTGDS | 0.09 | 8.2E − 01 | 0.07 | 1.0E + 00 | 0.37 | 5.7E − 04[a] | 0.03 | 1.0E + 00 |
| ABCG1 | − 0.07 | 8.2E − 01 | − 0.03 | 1.0E + 00 | 0.29 | 2.5E − 03[a] | − 0.02 | 1.0E + 00 |

Table presents the log(fold-change) before-after intervention and the false discovery rate (FDR). Genes considered specific for healthy individuals had FDR < 0.01 in healthy and FDR > 0.9 in metabolically impaired individuals (Obese/T2D). The top 5 up- and downregulated for aerobic and resistance were selected based on fold-change
[a]Significance at FDR < 0.05

MetaMEx interrogates transcriptomic response of skeletal muscle to exercise in obese, type 2 diabetic and/or people with the metabolic syndrome. Using a similar meta-analysis approach, we discovered an overrepresentation of pathways related to vasculature development and muscle organ development specifically induced by exercise training in metabolically impaired individuals. The adverse response to exercise training in metabolically impaired individuals was also driven by a subset of genes that are not annotated to any defined gene ontology pathway. These targets open new avenues of research and therapeutic perspectives. Although transcriptomic studies allow a deep and precise characterization of mRNA changes, other biological responses are relevant for the adaptive response of muscle to exercise, including phosphorylation cascades activating metabolic enzymes such as Akt and AMPK[32,33], or alterations in DNA structure leading to the establishment of a new steady state through epigenetic modifications[34]. Comprehensive multi-omics analyses will be required to reveal the full spectrum of exercise-induced adaptations of skeletal muscle in healthy and metabolically impaired individuals.

In addition to the wealth of scientific data on molecular transducers of the acute and chronic response to different exercise modalities, our analysis has overcome several limitations of the available studies designed to assess the skeletal muscle transcriptome. Furthermore, we highlight the need for additional studies of female participants, greater diversity in exercise types and modalities, and better reporting of experimental conditions such as timings of biopsy collection. These, and other limitations hinder researchers in their efforts to mine transcriptomic data, therefore underutilizing funding investments. Our study resonates with the ongoing Molecular Transducers of Physical Activity Consortium (MoTrPAC), which aims to characterize the human response to exercise depending on age, sex, body composition, fitness level, and exposure to exercise[35]. Approaches combining extensive human studies like MoTrPAC with meta-analyses methods like MetaMEx will be crucial to optimize, implement and coordinate omics research and open new exercise-based therapeutic perspectives.

## Methods

**Publicly available datasets**. Repositories were screened for human studies of inactivity (bed rest and unloading), and resistance and aerobic exercise, including acute exercise and exercise training studies. GSE42507, GSE5792, and GSE74194 were excluded because of the absence of Pre-exercise control groups. GSE43856 and GSE27285 were used as a baseline analysis for GSE44818, since these three studies were performed on the same platforms (Illumina HumanHT-12 V3.0 expression beadchip) and enrolled subjects of similar age, BMI and VO2max.

Studies without annotation or protocols that consisted of concurrent aerobic and resistance exercise were excluded. The final meta-analysis included a total of 66 studies from various gene arrays and RNAseq. Studies were annotated by skeletal muscle type (*vastus lateralis*, *biceps brachii* or *quadriceps femoris*) sex, age, weight, fitness level and diseases state and for healthy (HLY: BMI < 25, no metabolic disease) and metabolically impaired individuals (MTI: BMI > 25 and/or type 2 diabetes (T2D)). The clinical characteristics of the studies are presented in Table 1 and described in full details in Supplementary Data 1.

**MetaMEx—Individual array analysis**. The workflow is presented in Fig. 1. Whenever available, raw data was downloaded and re-analyzed using packages from the Bioconductor consortium. Robust multiarray averaging normalization was used for Affymetrix arrays[36]. For other types of arrays, and when raw data was not available, simple quantile normalization was used. RNA sequencing was analyzed from raw counts using the DESeq2 package[37]. Arrays were annotated with the ENSEMBL database and samples labeled according to the various study groups. Within each array and each study group, differentially expressed genes were identified using standard statistics. Mean, variance, fold-change, p-value, FDR (Benjamini–Hochberg) and 95% confidence intervals were calculated using the limma and matrixStats packages[38].

**MetaMEx—PCA and correlation analysis**. Genes with more than 10% missing values were excluded and the remaining missing values replaced with nearest neighbor averaging imputation. The log2(fold-change) for each study was then used to plot the principal component analysis (PCA) and the correlation matrix.

**MetaMEx—Meta-analysis**. For each gene, mean, variance and n size were used to fit a random effect model to the data. The restricted maximum-likelihood (REML) method was applied to the data using the R package metaphor[39]. The obtained p-values were adjusted using the Benjamini–Hochberg method.

**MetaMEx—Gene ontology and pathway analyses**. Gene ontology analysis was carried out in cluster profiler[40] on genes that passed an FDR < 0.01. KEGG pathways were used to compute a list of genes for enzymes involved in each gene set. For each protein in the pathway, an average of the fold-change of all the genes coding for that specific protein complex was performed.

**MetaMEx online tool**. For an accessible interrogation of the MetaMEx database, a web application was generated using R Shiny (https://shiny.rstudio.com). For each single gene, the application displays an output for the different types of exercise and inactivity protocols. For each study, the log2(fold-change), individual false discovery rate, and the 95% confidence interval is displayed. The bottom line of each graph shows the meta-analysis score. The app was programed for an easy selection of the studies and to allow the user to manually exclude or include specific sex, age or disease groups. Code is available at https://github.com/NicoPillon/MetaMEx.

**Validation cohorts**. The samples were obtained at Victoria University, Melbourne, Australia. Approval for all the experimental protocols and the study's procedures, which conformed to the standards set by the latest revision of the Declaration of Helsinki, was granted by the Victoria University Human Research Ethics Committee. Aerobic studies were previously published[41,42] and both studies included eight lean healthy men undergoing training aerobic exercise (age 20 years; VO2max 45.1 ml × min⁻¹ × kg⁻¹) or acute aerobic exercise (age 21 years; VO2max 46.7 ml ×

$min^{-1} \times kg^{-1}$). The resistance study included eight healthy men (age 27.4 years; $VO_2max$ 44.2 ml $\times$ min$^{-1} \times$ kg$^{-1}$) undergoing 8 weeks of resistance training. Acute resistance exercise: following a standardized warm-up (one set each of 5, then three repetitions at 50% and 60% 1-RM, respectively), participants performed 6 × 10 leg press repetitions at 70% 1-RM, separated by 2-min rest periods, on a plate-loaded 45° incline leg press (Hammer Strength Linear, Schiller Park, IL). Muscle was sampled immediately pre and 3 h post-exercise. Resistance training: For the remainder of the experimental training week, participants completed two more training days as described above. Following the first experimental week, the participants continued to train 3 days per week for 8 weeks. The resistance training program involved leg press, bench press, leg extension, seated row, and leg curl exercises (sessions 1 and 3), and leg press, dumbbell press, lat. pulldown, dumbbell lunges and leg curl exercises (session 2). The training intensity and volume progressed from three sets (weeks 2–5) to four sets (weeks 6–9) of 12- to 6-RM for each exercise, with 2-min rest between sets. Warm-up sets were performed prior to the first two exercises of each session (Session 1: leg press and bench press; Session 2: leg press and dumbbell chest press). At least 48 h after the last training session, a final resting biopsy was taken. Muscle samples (~180 ± 50 mg) were immediately frozen in liquid nitrogen, and stored at −80 °C until analysis.

**Primary human skeletal muscle cells**. Primary cells were isolated from *vastus lateralis* skeletal muscle biopsies derived from healthy male volunteers[43]. Myoblasts were let to proliferate in growth medium (F12/DMEM, 25 mM glucose, 20% FBS, 1% penicillin-streptomycin [Anti-anti, Thermo Fisher Scientific, Stockholm, Sweden]). Differentiation was induced in DMEM/M199 media containing HEPES (0.02 M; Invitrogen), zinc sulfate (0.03 µg/mL), vitamin $B_{12}$ (1.4 µg/mL; Sigma-Aldrich), insulin (10 µg/mL; Actrapid; Novo Nordisk), apo-transferrin (100 µg/mL; BBI Solutions), and 0.5% FBS and 1% penicillin-streptomycin [Anti-anti, Thermo Fisher Scientific, Stockholm, Sweden]). After 4 days, cells were switched to post-fusion medium containing DMEM/M199, HEPES, zinc sulfate, vitamin $B_{12}$, and 0.5% FBS and 1% penicillin-streptomycin [Anti-anti, Thermo Fisher Scientific, Stockholm, Sweden]).

**NR4A3 silencing**. Six days after inducing differentiation, cells were transfected with 10 nM of either silencer select Negative control No. 2 (no. 4390847) or validated silencer select siRNA s15542 to target *NR4A3* (Life Technologies, Foster City, CA). Transfections were performed in OptiMEM reduced serum media with Lipofectamine RNAiMAX transfection reagent (Invitrogen, Carlsbad, CA). Cells were exposed to two separate 5-h transfection periods separated by ~48 h. Two days after the final transfection, silencing efficiency was estimated using qPCR and Western blot and metabolic studies conducted as described below.

**Oxygen consumption and lactate production (Seahorse)**. To assess mitochondrial function and glycolytic rate of primary human skeletal myotubes, cells were subjected to a Seahorse XF Mito Stress Test using the manufacturer's instructions (Agilent, Santa Clara, CA). Cells were seeded at 30,000 per well in 24-well, assay-specific plates. Wells were washed and differentiation media was added the day after seeding. After differentiation, oxygen consumption rates (OCR) and extracellular acidification rates (ECAR) were measured at three timepoints under unstimulated conditions, then after treatment with 1 µM oligomycin, 2 µM carbonyl cyanide-4-(trifluoromethoxy)phenylhydrazone (FCCP), and 0.75 µM rotenone + antimycin A.

**Electrical pulse stimulation (EPS)**. Fully differentiated myotubes grown in 6-well plates were exposed to EPS using the C-Pace EP Culture Pacer (IonOptix, MA). Cells were washed with PBS and 2 mL of postfusion medium containing 5 mM glucose was added per well. Cells were pulsed according to two different protocols. Acute exposure was 40 V, 2 ms pulse duration, 1 Hz for up to 5 h.

**Electrical pulse stimulated glucose uptake**. Human primary myotubes transfected by either a siRNA against NR4A3 or a scrambled siRNA were subjected to electrical pulse stimulation (C-Pace EP, Ionoptix) at 40 V, 2 ms, 1 Hz during 3 h in low glucose serum-free DMEM. After 2 h stimulation, a solution of 1 mCi/mL 2-[1,2-³H]deoxy-D-glucose and 10 µmol/L unlabeled 2-deoxy-D-glucose was added to the medium. Glucose uptake was measured during the last hour of EPS. Cells were washed, lysed and subjected to scintillation counting.

**RNA extraction and analysis**. Cultured muscle cells were lysed and RNA was extracted using the E.Z.N.A Total RNA Kit (Omega Bio-tek, Norcross, GA) and concentration was determined through spectrophotometry. All equipment, software and reagents for performing the reverse transcription and qPCR were from Thermo Fisher Scientific. cDNA synthesis was performed from ~1 µg of RNA using random hexamers and the High Capacity cDNA Reverse Transcription kit according to manufacturer's instructions. We performed the qPCR using a StepOne Plus system with TaqMan Fast Universal PCR Master Mix and TaqMan probes (Supplementary Table 1). Validation of exercise-responsive genes was done using Custom TaqMan Array Plates (Thermo Fisher Scientific) with pre-designed probes (references available on request). Custom array plates were run on a

QuantStudio 7 Flex. Quantitative PCR was performed for 40 cycles (95 °C for 1 s, 60 °C for 20 s). Threshold cycle (Ct) values were determined with StepOne software (version 2.3) and the relative gene expression was calculated by the comparative ΔΔCt method.

**Protein extraction and immunoblot analysis**. Cells were lysed in homogenization buffer (137 mM NaCl, 2.7 mM KCl, 1 mM MgCl$_2$, 0.5 mM Na$_3$VO$_4$, 1% Triton X-100, 10% glycerol, 20 mM Tris at pH 7.8, 10 mM NaF, 1 mM EDTA, 0.2 mM PMSF, and 1% protease inhibitor cocktail (Merck, Darmstadt, Germany). Homogenates were rotated for 40 min at 4 °C and subjected to centrifugation (10,000 × g for 10 min at 4 °C). Protein content of the supernatants was measured by BCA protein assay kit (Pierce Biotechnology, Rockford, IL). Samples were prepared for SDS-PAGE with Laemmli buffer (60 mM Tris at pH 6.8, 2% w/v SDS, 10% v/v glycerol, 0.01% w/v bromophenol blue, 1.25% v/v β-mercaptoethanol). Equal amounts of protein were loaded and separated on Criterion XT Bis-Tris Gels (Bio-Rad, Hercules, CA) and transferred to polyvinylidene fluoride membranes (Merck, Darmstadt, Germany). Membranes were then stained with Ponceau S to confirm the quality of the transfer and equal loading of samples. Membranes were blocked with 5% non-fat milk in Tris-buffered saline with Tween-20 ((TBST); 20 mM tris·HCl at pH 7.6, 137 mM NaCl, 0.02% Tween-20) for 1 h at room temperature, and subsequently incubated overnight at 4 °C with primary antibodies diluted in TBS with 0.1% w/v bovine serum albumin and 0.1% w/v NaN$_3$. Antibodies were directed to NR4A3/NOR1 (Dilution 1:1000, NBP2-46246, Novus Biologicals) and mitochondrial complexes (Dilution 1:1000, Total OXPHOS Human WB Antibody Cocktail, Abcam Cat# ab110411, RRID:AB_2756818). Membranes were washed with TBST and incubated with species-appropriate horseradish peroxidase-conjugated secondary antibody (1:25,000 in TBST with 5% non-fat milk). Proteins were then visualized by enhanced chemiluminescence (Amersham ECL Western Blotting Detection Reagent, Little Chalfont, UK). Protein content was quantified by densitometry (QuantityOne, Bio-Rad, Hercules, CA).

**Statistics**. Data are presented as mean ± SEM. Normality was tested using Shapiro-Wilk test. Differences in mRNA expression between control and NR4A3 silencing were analyzed using independent Student $t$ tests. Results from the Seahorse, western blot of mitochondrial complexes and glucose uptake were analyzed using repeated-measures ANOVA (silencing/treatment) followed by a Sidak post hoc test. Comparisons were considered statistically significant at $P < 0.05$. Analyses were performed using either R 3.5.2 (www.r-project.org) or GraphPad Prism 8.1 software (GraphPad Software Inc.).

**Reporting summary**. Further information on research design is available in the Nature Research Reporting Summary linked to this article.

## Data availability
Original studies used for the meta-analysis are publicly available on the GEO repository https://www.ncbi.nlm.nih.gov/geo. The curated database (MetaMEx) generated during the current study is available at www.metamex.eu. The source data underlying all figures are provided as a Source Data file.

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

## Acknowledgements

The authors are supported by grants from the Novo Nordisk Foundation (NNF14OC0011493, NNF17OC0030088 and NNF14OC0009941), Swedish Diabetes Foundation (DIA2018-357, DIA2018-336), Swedish Research Council (2015-00165, 2018-02389), the Strategic Research Program in Diabetes at Karolinska Institutet (2009-1068), the Stockholm County Council (SLL20150517, SLL20170159), the Swedish Research Council for Sport Science (P2018-0097), and the EFSD European Research Programme on New Targets for Type 2 Diabetes supported by an educational research grant from MSD. L.D. was supported by a Novo Nordisk postdoctoral fellowship run in partnership with Karolinska Institutet. B.M.G. was supported by a fellowship from the Wenner-Gren Foundation (Sweden). N.J.P. was supported by an Individual Fellowship from the Marie Skłodowska-Curie Actions (European Commission, 704978, 675610) and grants from the Sigurd och Elsa Goljes Minne and Lars Hiertas Minne Foundations (Sweden). D.J.B. was supported by the ANZ Mason Foundation and Australian Research Council Discovery Program (ARC DP140104165). Additional support was received from the Novo Nordisk Foundation Center for Basic Metabolic Research at the University of Copenhagen (NNF18CC0034900) (to J.R.Z.). We thank Dr. Nanjiang Shu from National Bioinformatics Infrastructure Sweden (NBIS) for setting up the web-server. We also thank EGI federated cloud for providing the computer resource for hosting the web-server. We acknowledge the Beta Cell in-vivo Imaging/Extracellular Flux Analysis core facility supported by the Strategic Research Program (SRP) in Diabetes for the usage of the Seahorse flux analyzer.

## Author contributions

Conceptualization: N.J.P. and J.R.Z. Methodology: N.J.P., B.M.G., L.D., J.A.S., L.S.P., A.K. and J.R.Z. Investigation: N.J.P., B.M.G., L.D., J.A.S., L.S.P., J.B. and D.J.B. Formal analysis: N.J.P., B.M.G., L.D., J.A.S., L.S.P., J.B. and D.J.B. Resources: D.J.B., A.K. and J.R.Z. Software: N.J.P. Writing—Original draft: N.J.P., B.M.G. and J.R.Z. Writing—Review & editing: N.J.P., B.M.G., L.D., J.A.S., L.S.P., J.B., D.J.B., A.K. and J.R.Z. Funding acquisition: B.M.G., L.D., N.J.P., A.K. and J.R.Z.

## Competing interests

The authors declare no competing interests.
