## [Peer Review File · Nature Communications]

Reviewers' Comments:

Reviewer #1:

Remarks to the Author:

In their manuscript entitled „Transcriptomic profiling of skeletal muscle adaptations to exercise and inactivity“ Pillon et al. describe a novel online tool to explore changes in gene expression in skeletal muscle due to exercise or inactivity. The authors integrated 64 publicly available datasets using a meta-analysis approach. Using this method, they identified NR4A3 as one of the most exercise- and inactivity responsive genes. For validation purposes they used a variety of biomolecular methods to demonstrate that this gene may play a role in the metabolic response to contraction. Their approach is very innovative and is based on an enormous amount of work. They created an important resource of knowledge in exercise research that is freely available for everyone. However, I have several concerns regarding the interpretation of their results. I am specifically interested why the authors did not perform an individual participant meta-analysis. If I am understanding their manuscript correctly they analyzed each array but the phenotypic description of arrays from the same study only included summarized data from the participant description in the previously published studies. This leads to an unnecessary loss of information and significantly weakens the ability for appropriate interpretation of their results. Further, Pillon et al. explore transcriptomic changes of physical (in)activity as if they were on some sort of continuum with one being one end and one being on the other end. But this is simply not the case (1). Exercise and inactivity reside in different mechanistic planes, and are not merely mirror images of each other. Even though the online tool developed by Pillon and colleagues may provide novel insight into transcripts associated with one or the other type of exercise training, optimal therapies and preventive strategies require knowledge of causal mechanisms. Thus, it is important to understand that some of the mechanisms by which inactivity causes chronic diseases differ from mechanisms by which exercise acts as primary prevention of same diseases. This is currently not discussed on the manuscript.

Minor comments:

The author do not provide any n's for the biomolecular analysis of NR4A3.

References:

1. Booth FW, Roberts CK, and Laye MJ. Lack of exercise is a major cause of chronic diseases. *Comprehensive Physiology* 2: 1143-1211, 2012.

Reviewer #2:

Remarks to the Author:

In this paper, Pillon et al. conducted a series of meta analyses to profile the skeletal muscle transcriptome to exercise and inactivity using 64 published datasets. Gene ontology and pathway analyses reveal selective pathways activated by inactivity, aerobic versus resistance and acute versus chronic exercise training. The authors also identified NR4A3 as one of the most exercise- and inactivity-responsive genes, and established a role for this nuclear receptor in mediating the metabolic responses to exercise-like stimuli in vitro. Some comments:

1. Overall, the authors have done a remarkable job collecting extensive amount of transcriptomics data from difference sources and painstakingly conducted a comprehensive meta analyses, looking at various levels of genes and pathways. The web-based software tool, MetaMEx, seems very powerful and easy-to-use. But I found most of the results are descriptive and unremarkable. The biggest finding is the NR4A3 gene. But it seems that there have been several reports in the literature linking it to response to exercise in skeletal muscle already. Given these, I am not sure if there is sufficient novel findings resulted from the meta analysis.
2. Figure 2. At the end of the figure legend, it is stated that “*Insufficient number of studies to appropriately calculate a Meta-analysis score in healthy individuals.” However, I did not see any “*” in this figure. Are the authors referring to rows with “NA”?

3. Figure 4. (B) how are the pathways ordered? (C)-(F), what does the height of the box indicate? I have no idea what is the point the authors are trying to get across.
4. Figure 5. (C), what does "****" indicate?
5. What is the unique feature of MetaMEx that differentiate it from other meta analysis tools?

Reviewer #3:

Remarks to the Author:

To the authors

The paper by Pillon et al recognizes the underutilization of publicly available data derived from large-scale genomic studies towards the effect of a range of physical (in)activity interventions in healthy and metabolically compromised individuals on the transcriptomic profile in skeletal muscle. The paper not only recognizes the underutilization of these data but also presents an in home developed, publicly available web based tool (www.metamax.eu) that permits a meta-analysis of the currently available data sets than can be readily expanded with data-sets to come. Without any doubt, this is a highly valuable tool, which will help us to better understand and possibly predict the effects of physical activity. With this tool, the authors identified NR4A3 as the most exercise and inactivity responsive gene and choose to study this particular gene in more detail.

Major issues

As it reads now the paper is (too) ambiguous; it serves two completely distinct goals (presenting and positioning the metamax tool and identification of NR4A3 as a novel exercise and inactivity gene). It does not become clear from the study as to why the authors chose to cherry-pick NR4A3 for follow-up studies, and not e.g., MAFF or GADD45G. In my opinion, it would be better to split the paper into a paper presenting and positioning the metamax toolbox (for which I doubt if Nature Communications is the journal with the best fit) and another paper on NR4A3, that really would need additional work before it is suitable as a standalone paper in a high-impact journal.

Minor issues:

- In the abstract, it is correctly mentioned that exercise may prevent and ameliorate metabolic diseases and slow secondary ageing. The abstract also states that the molecular mechanisms are not fully understood. A statement that is also correct, but that also suggests that with the tool presented, these issues will be solved/circumvented, which is not the case. The metamax tool does not (yet?) link to disease outcomes etc. so please rephrase this section.
- It is mentioned that 64 published data-sets have been used. In figure 2, however, 'only' 63 studies are being mentioned.
- The figure 2 caption is referring to an asterix (insufficient number of studies), which I do not seem to trace in the figure.
- On page 3 it is mentioned that BMI was 'increased' in metabolically compromised vs healthy individuals. This suggests an intervention effect, whereas only cross-sectional data are available. Please change 'increased' into 'higher'.
- In the results section describing figure 2 it is mentioned that PGC1a is upregulated 2.4 fold after acute aerobic exercise, with a REML of 1.15 however, the fold change would be $2^{1.15}=2.219$ fold increase, right?
- There seems to be a mismatch between the numbers in figure 2 in the manuscript and the same data that can be found at www.metamax.eu, even though the original studies included seem identical. Please check and explain or make consistent.

I would like to end by expressing my greatest appreciation for the toolbox developed and all the hard and well thought of work that has been performed and making the toolbox publicly available. Clearly, the authors highly value and appreciate the need to report these essential details. This notion is underscored by presenting (in supplemental data) material that often remains under reported (gender, diet prior to biopsy, site of biopsy, duration after final exercise session etc.).

Reviewer #1 (Remarks to the Author):

In their manuscript entitled „Transcriptomic profiling of skeletal muscle adaptations to exercise and inactivity“ Pillon et al. describe a novel online tool to explore changes in gene expression in skeletal muscle due to exercise or inactivity. The authors integrated 64 publicly available datasets using a meta-analysis approach. Using this method, they identified NR4A3 as one of the most exercise- and inactivity responsive genes. For validation purposes they used a variety of biomolecular methods to demonstrate that this gene may play a role in the metabolic response to contraction. Their approach is very innovative and is based on an enormous amount of work. They created an important resource of knowledge in exercise research that is freely available for everyone. However, I have several concerns regarding the interpretation of their results.

RESPONSE: We appreciate the reviewer’s recognition of the amount of work we put into the meta-analysis. Please find below our point-by-point response to your comments below.

I am specifically interested why the authors did not perform an individual participant meta-analysis. If I am understanding their manuscript correctly, they analyzed each array but the phenotypic description of arrays from the same study only included summarized data from the participant description in the previously published studies. This leads to an unnecessary loss of information and significantly weakens the ability for appropriate interpretation of their results.

RESPONSE: Unfortunately, phenotypic/clinical data is often only available in the published papers in a summary table with means and Sd. In most publications, the authors do not report individual values and/or do not annotate the arrays in a way that one can match the data to individual donors. We agree with the reviewer that information is lost in this process, which highlights the need for better reporting of the data associated with publications. To be able to include all studies, we had to use the summarized clinical data that were available in the published papers.

Further, Pillon et al. explore transcriptomic changes of physical (in)activity as if they were on some sort of continuum with one being one end and one being on the other end. But this is simply not the case (1). Exercise and inactivity reside in different mechanistic planes, and are not merely mirror images of each other. Even though the online tool developed by Pillon and colleagues may provide novel insight into transcripts associated with one or the other type of exercise training, optimal therapies and preventive strategies require knowledge of causal mechanisms. Thus, it is important to understand that some of the mechanisms by which inactivity causes chronic diseases differ from mechanisms by which exercise acts as primary prevention of same diseases. This is currently not discussed on the manuscript.

RESPONSE: We completely agree with your comments. We have removed the word “conversely” of the introduction (line 31) and added a section in the discussion that highlights the point you raise regarding that it is important to understand that some of the mechanisms by which inactivity cause chronic diseases differ from mechanisms by which exercise acts primary prevention of the same diseases (lines 285-292).

Minor comments:

The authors do not provide any n’s for the biomolecular analysis of NR4A3.

RESPONSE: All n sizes are written in the figure legend and data are presented as individual points.

References:

1. Booth FW, Roberts CK, and Laye MJ. Lack of exercise is a major cause of chronic diseases. *Comprehensive Physiology* 2: 1143-1211, 2012.

Reviewer #2 (Remarks to the Author):

In this paper, Pillon et al. conducted a series of meta analyses to profile the skeletal muscle transcriptome to exercise and inactivity using 64 published datasets. Gene ontology and pathway analyses reveal selective pathways activated by inactivity, aerobic versus resistance and acute versus chronic exercise training. The authors also identified NR4A3 as one of the most exercise- and inactivity-responsive genes and established a role for this nuclear receptor in mediating the metabolic responses to exercise-like stimuli in vitro. Some comments:

1. Overall, the authors have done a remarkable job collecting extensive amount of transcriptomics data from difference sources and painstakingly conducted a comprehensive meta analyses, looking at various levels of genes and pathways. The web-based software tool, MetaMEx, seems very powerful and easy-to-use. But I found most of the results are descriptive and unremarkable. The biggest finding is the NR4A3 gene. But it seems that there have been several reports in the literature linking it to response to exercise in skeletal muscle already. Given these, I am not sure if there is sufficient novel findings resulted from the meta-analysis.

RESPONSE: A meta-analysis re-processes published data, so the potential for novel discoveries is related to the gain in statistical power. Our study is the first of its kind to pool and use meta-analysis statistical methods on full transcriptomic datasets to compare different protocols of exercise acute, training and inactivity. We unravel subtle differences between aerobic versus resistance exercise and identify subsets of genes responsive to exercise only in obese/Type 2 diabetic individuals, which potentially opens novel avenues of research. As the reviewer points out, we provide the most comprehensive database to date on the skeletal muscle response to exercise with 66 curated studies and an online tool to access the data, which is a unique contribution to the field.

In addition, our revised manuscript now includes novel data in human primary muscle cells on the role of 4 exercise-responsive targets (DNAJA4, KLHL40, NR4A3 and VGLL2) in the regulation of inactivity-responsive genes and the response to contraction (Fig 5 and Supp Fig 2).

2. Figure 2. At the end of the figure legend, it is stated that “*Insufficient number of studies to appropriately calculate a Meta-analysis score in healthy individuals.” However, I did not see any “*” in this figure. Are the authors referring to rows with “NA”?

RESPONSE: The asterisks were placed after the titles, but we realized from the reviewers’ comments that this was unclear. We rephrased the legend as: “In the case of HIIT training and combined exercise training protocols, the number of studies is insufficient to calculate meaningful meta-analysis statistics”.

3. Figure 4. (B) how are the pathways ordered? (C)-(F), what does the height of the box indicate? I have no idea what is the point the authors are trying to get across.

RESPONSE: The pathways were organized by decreasing significance in the order of acute aerobic, acute resistance, inactivity, training aerobic and training resistance exercise. We now provide a curated figure to clarify this point.

The plots in C-F shows the $\log_2(\text{fold-change})$ for a given gene in a pathway. The total height of the bar considers all the fold-changes for all the genes in a pathway. For instance, measuring all the genes involved in mitochondrial respiration shows that all genes involved in mitochondrial complexes are decreased by inactivity. We have adjusted the figure legend to better describe how the figure was created (lines 764-767).

4. Figure 5. (C), what does “***” indicate?

RESPONSE: *** $p < 0.001$ has been added to the figure legend.

5. What is the unique feature of MetaMEx that differentiate it from other meta analysis tools?

RESPONSE: To the best of our knowledge, MetaMEx is the only available database that comprehensively pools, normalizes and compares all exercise and inactivity transcriptomic studies. We are not aware of any other tool that allows for a meta-analysis of 66 transcriptomic studies at once, and the MetaMEx online app is a unique open-source tool for interrogating the database.

Reviewer #3 (Remarks to the Author):

To the authors

The paper by Pillon et al recognizes the underutilization of publicly available data derived from large-scale genomic studies towards the effect of a range of physical (in)activity interventions in healthy and metabolically compromised individuals on the transcriptomic profile in skeletal muscle. The paper not only recognizes the underutilization of these data but also presents an in home developed, publicly available web based tool (www.metamax.eu) that permits a meta-analysis of the currently available data sets than can be readily expanded with data-sets to come. Without any doubt, this is a highly valuable tool, which will help us to better understand and possibly predict the effects of physical activity. With this tool, the authors identified NR4A3 as the most exercise and inactivity responsive gene and choose to study this particular gene in more detail.

RESPONSE: We thank the reviewer for the encouraging comments and recognizing the value of MetaMEx. We provide a point-by-point answer to your comments below.

Major issues

As it reads now the paper is (too) ambiguous; it serves two completely distinct goals (presenting and positioning the metamax tool and identification of NR4A3 as a novel exercise and inactivity gene). It does not become clear from the study as to why the authors chose to cherry-pick NR4A3 for follow-up studies, and not e.g., MAFF or GADD45G. In my opinion, it would be better to split the paper into a paper presenting and positioning the metamax toolbox (for which I doubt if Nature Communications is the journal with the best fit) and another paper on NR4A3, that really would need additional work before it is suitable as a standalone paper in a high-impact journal.

RESPONSE: We apologize if our rationale for choosing NR4A3 was unclear. Our filtering involved overlapping the datasets, selecting the most significant genes modified by acute aerobic and resistance exercise and identifying changes in gene expression common between exercise and inactivity. In particular, we were interested in genes regulated in an opposing manner (i.e. increased by exercise and reduced by inactivity). We identified 4 common genes: *DNAJA4*, *KLHL40*, *NR4A3*, *VGLL2* (Figure 4A).

Figure 4A. Selection pipeline of exercise and inactivity responsive genes. Genes significantly modified by acute aerobic and resistance exercise and inactivity were overlapped in a Venn Diagram.

We validated the four targets in human biopsies from independent cohorts and determined the role of these genes in regulating the response to electrical pulse stimulation in mouse C2C12 and primary human cells. We also tested whether silencing of these genes mimicked the response to inactivity as observed in MetaMEx. Based on our data and the literature, we focused our further validation efforts on *NR4A3*. We have now added supplemental data describing our analysis for the 4 targets and rephrased the manuscript accordingly (Figure 4 and Supplemental Figure 3).

Minor issues:

- In the abstract, it is correctly mentioned that exercise may prevent and ameliorate metabolic diseases and slow secondary ageing. The abstract also states that the molecular mechanisms are not fully understood. A statement that is also correct, but that also suggests that with the tool presented, these issues will be solved/circumvented, which is not the case. The metamex tool does not (yet?) link to disease outcomes etc. so please rephrase this section.

RESPONSE: The abstract has now been rephrased to be less ambiguous.

- It is mentioned that 64 published datasets have been used. In figure 2, however, ‘only’ 63 studies are being mentioned.

RESPONSE: The number of studies has been updated regularly and the most current database includes 66 studies. This has been corrected in both the text and figure legend.

- The figure 2 caption is referring to an asterisk (insufficient number of studies), which I do not seem to trace in the figure.

RESPONSE: The asterisks were placed after the title, but we realized from the reviewers’ comments that this was not ideal. We rephrased the legend as: “In the case of HIIT training and combined exercise training protocols, the low number of studies is insufficient to calculate meaningful meta-analysis statistics”.

- On page 3 it is mentioned that BMI was ‘increased’ in metabolically compromised vs healthy individuals. This suggests an intervention effect, whereas only cross-sectional data are available. Please change ‘increased’ into ‘higher’.

RESPONSE: This has been corrected.

- In the results section describing figure 2 it is mentioned that PGC1a is upregulated 2.4 fold after acute aerobic exercise, with a REML of 1.15 however, the fold change would be $2^{1.15}=2.219$ fold increase, right?

RESPONSE: The reviewer is correct; we apologize for this error. In the most recent database, PGC1A is increased by 1.227167 (log2), so it is now 2.3-fold. The result section has been updated accordingly.

- There seems to be a mismatch between the numbers in figure 2 in the manuscript and the same data that can be found at www.metamex.eu, even though the original studies included seem identical. Please check and explain or make consistent.

RESPONSE: This was due to a lag between the local and online databases, as they are not updated simultaneously. We have now updated both the manuscript and the online tool with the latest versions to correct these discrepancies. Note that the online tool is constantly updated with new data, while the data used for manuscript is limited to datasets that were included at the time of submission and eventual publication, so discrepancies will occur in the future and the online tool should be considered as the reference.

I would like to end by expressing my greatest appreciation for the toolbox developed and all the hard and well thought of work that has been performed and making the toolbox publicly available. Clearly, the authors highly value and appreciate the need to report these essential details. This notion is underscored by presenting (in supplemental data) material that often remains under reported (gender, diet prior to biopsy, site of biopsy, duration after final exercise session etc.).

RESPONSE: Many thanks for the supportive comments! The tables indeed contain all the information we could gather for all the studies. It is a unique resource, and we made sure that is available both as supplemental and in a user-friendly searchable table on www.metamex.eu

Reviewers' Comments:

Reviewer #1:

Remarks to the Author:

I congratulate the authors to a great manuscript.

The MetaMEx tool developed will provide exercise physiologists and researchers with great insight into the transcriptomic adaptations of skeletal muscle acute and chronic exercise as well as inactivity.

I only have minor concerns:

- Ln. 46: myriad OF
- Ln. 76: to should be of
- Ln. 115: please spell out acronym PCA at first use. Further, use spell out PCA in 230 and 369. Please be consistent.
- Ln. 163: please add commas before and after therefore
- Ln. 171: please change sentence structure to "genes coding for mitochondrial complexes"
- Ln. 426 and Ln. 428: please provide location and country for Invitrogen and Life Technologies
- Ln. 438: please spell out FCCP
- Ln. 455: please do not start a sentence with a small letter
- Ln. 467: This is the first time you put a space between 10 000 g. Prior this number were just written out. Please be consistent.
- Ln. 472: please provide city and country for Merck
- Ln. 480: Please spell out TBST
- Ln. 484: Please provide city and country for Biorad

Reviewer #2:

Remarks to the Author:

I have no further comment.

Reviewer #3:

Remarks to the Author:

To the authors

The authors satisfactorily addressed all of my concerns. I particularly appreciate the more elaborate part on as to why the NR4A3 gene was selected for analysis that is more detailed. I feel that the 'exercise-health community' will highly appreciate this valuable and important piece of work.

REVIEWERS' COMMENTS:

RESPONSE: We thank the three reviewers for their comments and recognition of the value of MetaMEx. We have addressed all of the minor comments listed below by Reviewer 1 in the revised manuscript.

Reviewer #1 (Remarks to the Author):

I congratulate the authors to a great manuscript. The MetaMEx tool developed will provide exercise physiologists and researchers with great insight into the transcriptomic adaptations of skeletal muscle acute and chronic exercise as well as inactivity.

I only have minor concerns:

- ln. 46: myriad OF
- ln. 76: to should be of
- ln. 115: please spell out acronym PCA at first use. Further, use spell out PCA in 230 and 369. Please be consistent.
- ln. 163: please add commas before and after therefore
- ln. 171: please change sentence structure to “genes coding for mitochondrial complexes”
- ln. 426 and ln. 428: please provide location and country for Invitrogen and Life Technologies
- ln. 438: please spell out FCCP
- ln. 455: please do not start a sentence with a small letter
- ln. 467: This is the first time you put a space between 10 000 g. Prior this number were just written out. Please be consistent.
- ln. 472: please provide city and country for Merck
- ln. 480: Please spell out TBST
- ln. 484: Please provide city and country for Biorad

Reviewer #2 (Remarks to the Author):

I have no further comment.

Reviewer #3 (Remarks to the Author):

The authors satisfactorily addressed all of my concerns. I particularly appreciate the more elaborate part on as to why the NR4A3 gene was selected for analysis that is more detailed. I feel that the ‘exercise-health community’ will highly appreciate this valuable and important piece of work.